# Early human fetal lung atlas reveals the temporal dynamics of epithelial cell plasticity

Henry Quach [1,2,10], Spencer Farrell [3,10], Ming Jia Michael Wu [1], Kayshani Kanagarajah[1,2], Joseph Wai-Hin Leung[4], Xiaoqiao Xu[5], Prajkta Kallurkar[5], Andrei L. Turinsky[5], Christine E. Bear[6], Felix Ratjen[7], Brian Kalish[4,8,9], Sidhartha Goyal[3], Theo J. Moraes[7] & Amy P. Wong [1,2] ✉

Studying human fetal lungs can inform how developmental defects and disease states alter the function of the lungs. Here, we sequenced >150,000 single cells from 19 healthy human pseudoglandular fetal lung tissues ranging between gestational weeks 10–19. We capture dynamic developmental trajectories from progenitor cells that express abundant levels of the cystic fibrosis conductance transmembrane regulator (*CFTR*). These cells give rise to multiple specialized epithelial cell types. Combined with spatial transcriptomics, we show temporal regulation of key signalling pathways that may drive the temporal and spatial emergence of specialized epithelial cells including ciliated and pulmonary neuroendocrine cells. Finally, we show that human pluripotent stem cell-derived fetal lung models contain *CFTR*-expressing progenitor cells that capture similar lineage developmental trajectories as identified in the native tissue. Overall, this study provides a comprehensive single-cell atlas of the developing human lung, outlining the temporal and spatial complexities of cell lineage development and benchmarks fetal lung cultures from human pluripotent stem cell differentiations to similar developmental window.

Single-cell technologies are redefining the molecular signatures of canonical cell types and identifying new cell types and cell states in organs and tissues[1,2]. Specifically, single-cell RNA sequencing (scRNA-seq) has revealed cellular transcriptomic heterogeneities and has enriched our understanding of how these cells contribute to development and disease. Single-cell atlases of the adult human pulmonary system have illuminated the roles of key cell types in homeostasis and

some diseases[3–7]. On the contrary, our understanding of the cells that form the developing fetal lung has largely been extrapolated from mouse studies[8,9]. Recently, there have been studies leveraging scRNA-seq to characterize the developing human fetal lung tissue[10–14]. Through single-cell studies, species-specific differences in lung cell types and cellular distributions have been identified, which can impact function and disease pathogenesis. Understanding the molecular and

[1]Program in Developmental and Stem Cell Biology, Hospital for Sick Children, Toronto, Ontario, Canada. [2]Department of Laboratory Medicine & Pathobiology, University of Toronto, Toronto, Ontario, Canada. [3]Department of Physics, University of Toronto, Toronto, Ontario, Canada. [4]Program in Neurosciences and Mental Health, Hospital for Sick Children, Toronto, Ontario, Canada. [5]Centre for Computational Medicine, Hospital for Sick Children, Toronto, Ontario, Canada. [6]Program in Molecular Medicine, Hospital for Sick Children, Toronto, Ontario, Canada. [7]Program in Translational Medicine, Hospital for Sick Children, Toronto, Ontario, Canada. [8]Department of Molecular Genetics, University of Toronto, Toronto, Ontario, Canada. [9]Division of Neonatology, Department of Paediatrics, Hospital for Sick Children, Toronto, Ontario, Canada. [10]These authors contributed equally: Henry Quach, Spencer Farrell. ✉e-mail: apwong@sickkids.ca

cellular networks that drive the formation of the respiratory system during fetal development may provide important insight into the regenerative processes during repair and potentially identify the developmental origins of postnatal lung disease and disorders.

Recent sequencing datasets from a small collection of normal human fetal lungs obtained from early to mid-gestation fetuses have highlighted some novel cell types and cell lineage trajectories[10,12] that are unique to human fetal development. Unfortunately, following up on these observations and their potential impact on post-natal lung function is hindered by a lack of relevant experimentally tractable models.

Human pluripotent stem cells (hPSCs) are becoming a resourceful tool to generate organoids or in vitro tissue mimetics to study fundamental mechanisms of human cell lineage development. Directed differentiation protocols towards mature lung epithelia[15–17] have achieved robust differentiations by recapitulating developmental milestones, and have been used to discover cell therapies for lung diseases such as cystic fibrosis[18–20]. However, to precisely model the native tissue, a comprehensive understanding of the cells that exist and how they develop requires in-depth analysis of the native tissue. Understanding the developmental trajectories and pathways driving these cell lineage formations will improve current hPSC differentiation protocols to generate bona fide cell types.

Here, we present a comprehensive single-cell atlas of freshly isolated human fetal lungs from 19 independent lung samples spanning gestational weeks 10–19 and capturing >150,000 cells. After dataset integration of all the samples, we identified 58 cell types/cell states in the fetal lungs during this period of gestation. Given the large number of cells sequenced, our dataset revealed important dynamic epithelial cell plasticity and lineage trajectories. To determine the ancestral origins and cell lineage relationships, we leveraged our previously established modified RNA velocity prediction tool, LatentVelo[21], followed by partition-based graph abstraction (PAGA) visualization, and Slingshot[22] to identify classical trajectories in the epithelial, immune, stromal, and endothelial compartments and more importantly trajectories with cellular transitions from unexpected cellular origins in the developing epithelium. We observed canonical cell types previously found in both adult human lungs[4,7] and embryonic mouse lungs[9], and also identified 4 airway epithelial progenitor cell subtypes that express abundantly high levels of *CFTR*. Importantly, we revealed the dynamic nature of how these *CFTR*-expressing epithelial progenitors contribute to the specialized cell types in the developing airways including pulmonary neuroendocrine (PNEC), multi-ciliated, basal and club cell lineages. In combination with high-resolution spatial transcriptomics, we revealed the temporal changes in intercellular signaling between sending and receiving cells that may contribute to dynamic cell fates, specifically through the *SCGB3A2 + SFTPB + CFTR +* (triple positive, or TP) cells, a cell population recently identified[13]. Finally, we showed that hPSC-derived fetal lung epithelial models contained *CFTR*-expressing epithelial cell progenitor cells and the differentiation also captured the same developmental trajectories observed in the native fetal lung epithelium. The benchmarking of hPSC differentiations supports the use of these fetal lung models for future live lineage tracing and functional analyses of their role in airway development.

## Results

### Cell composition of the human fetal lungs

A total of 19 freshly isolated fetal lung tissues from elective pregnancy terminations ranging from gestational weeks (GW) 10–19 were collected and processed for 3′ single-cell RNA sequencing (scRNA-seq, 10X Genomics). One tissue was sampled twice from different regions of the lung to ensure even capture and representation of the lungs (denoted with an asterisk in Supplementary Table 1). A total of 170,256 single cells were sequenced at ~60,000 read depth per cell. FASTQ files were generated, and the reads were aligned with the

human reference genome hg38 (GRCh38) (Supplementary Fig. 1A). After the removal of low-quality cells, dead cells, and doublets, 156,698 cells were analyzed. After normalization and dimensionality reduction, reciprocal principal component analysis (rPCA) was used to identify anchors for integration to generate the fetal lung dataset. Twelve male and seven female lungs were collected based on the expression of the male sex-determining genes, *SRY* and *DDX3Y*, as well as the *XIST* (Fig. 1A). Leiden clustering informed by Clustree[23] was used to unbiasedly define the number of cell clusters (Supplementary Fig. 1B). Differentially expressed genes (DEG) based on non-parametric Wilcoxon rank sum test were used to annotate the cell populations which were broadly classified into 5 uniquely different cell populations (Fig. 1B and Supplementary Data 1) designated as stromal (*N* = 98,166; *COL1A1, COL1A2*), epithelial (*N* = 16,068; *EPCAM*), endothelial (*N* = 13,376; *PECAM1*), immune (*N* = 16,258; *CD74, CD3, NKG7*), and schwann cells (*N* = 512; *NRXN1, S100B, PLP1*) (Fig. 1C). Further sub-clustering of each main cell population was informed by Clustree and DEG which resulted in a total of 58 distinct cell types/states in the developing lungs (Fig. 1D). Assessing the cell proportions over gestational time showed stromal cells constituted the largest proportion of cells throughout the 10 weeks of development (Fig. 1E).

We performed an integrated analysis to identify similarities and differences between our dataset and three recently published datasets encompassing similar developmental lung tissue[10–12]. An integrated UMAP overlay of our dataset (in red, Fig. 1F) with the published datasets by Cao et al., He et al., and Sountoulidis et al., (combined in gray) identified 54 of the 58 cell types (~ 93%) were common amongst all the datasets (Supplementary Table 2 and Supplementary Fig. 1C). However, 4 unique cell types were exclusively captured in our dataset and include basal cells, mature ciliated cells, and submucosal gland (SMG) basal cells in the epithelial cluster and monocyte 2 from the immune cluster. This may reflect the differences in cluster annotations, cluster resolution, and differences in DEG associated with each cell type in the datasets and/or sampling error. Additional comparisons between our dataset with each published fetal atlases are detailed in Supplementary Fig. 2 and Supplementary Note. 1. Future efforts to integrate all the datasets and harmonize cluster annotations will strengthen the collectively rich resource of fetal lung atlases available.

Herein, we will describe the epithelial and stromal cell compartments and additional analyses of the fetal lung endothelium in Supplementary Fig. 3 and Supplementary Note. 2, and immune cells in Supplementary Fig. 4 and Supplementary Note. 3.

We next compared our fetal lung dataset to a published human adult lung single-cell atlas using the MapQuery function in Seurat to identify cells of similar biological states of the adult lung[7] onto our reference fetal lung dataset. As the percentage of anchors to query cells was less than 15% (~ 4%), only high confidence prediction scores (greater than 0.9) were used to predict cell identity. This resulted in 83% of fetal lung cell types captured in the adult lung (Supplementary Fig. 5A). Notable cell types that were not captured included tip cells and *NKX2-1 + SOX9 + CFTR +* cells. We subsequently reversed the reference and query datasets and found only 56% of adult lung cell types captured in our fetal lung dataset (Supplementary Fig. 5B). Cell types that were not captured included alveolar-associated cell types (i.e., alveolar fibroblasts and alveolar macrophages), nasal cells (goblet (nasal) and multiciliated (nasal)), and ionocytes.

We also compared our dataset to a previous mouse embryonic single lung cell atlas by Negretti et al.[9] and found 62% common cell types (Supplementary Fig. 5C). It should be noted that the mouse lung dataset by Negretti et al captures tissues from embryonic day 12 and 15 (similar to pseudoglandular lung tissues in human). Therefore, the large number of cells not shared between our fetal lung dataset with the mouse embryonic lung datasets may be explained by the differences in developmental stage of the tissue as well as species differences.

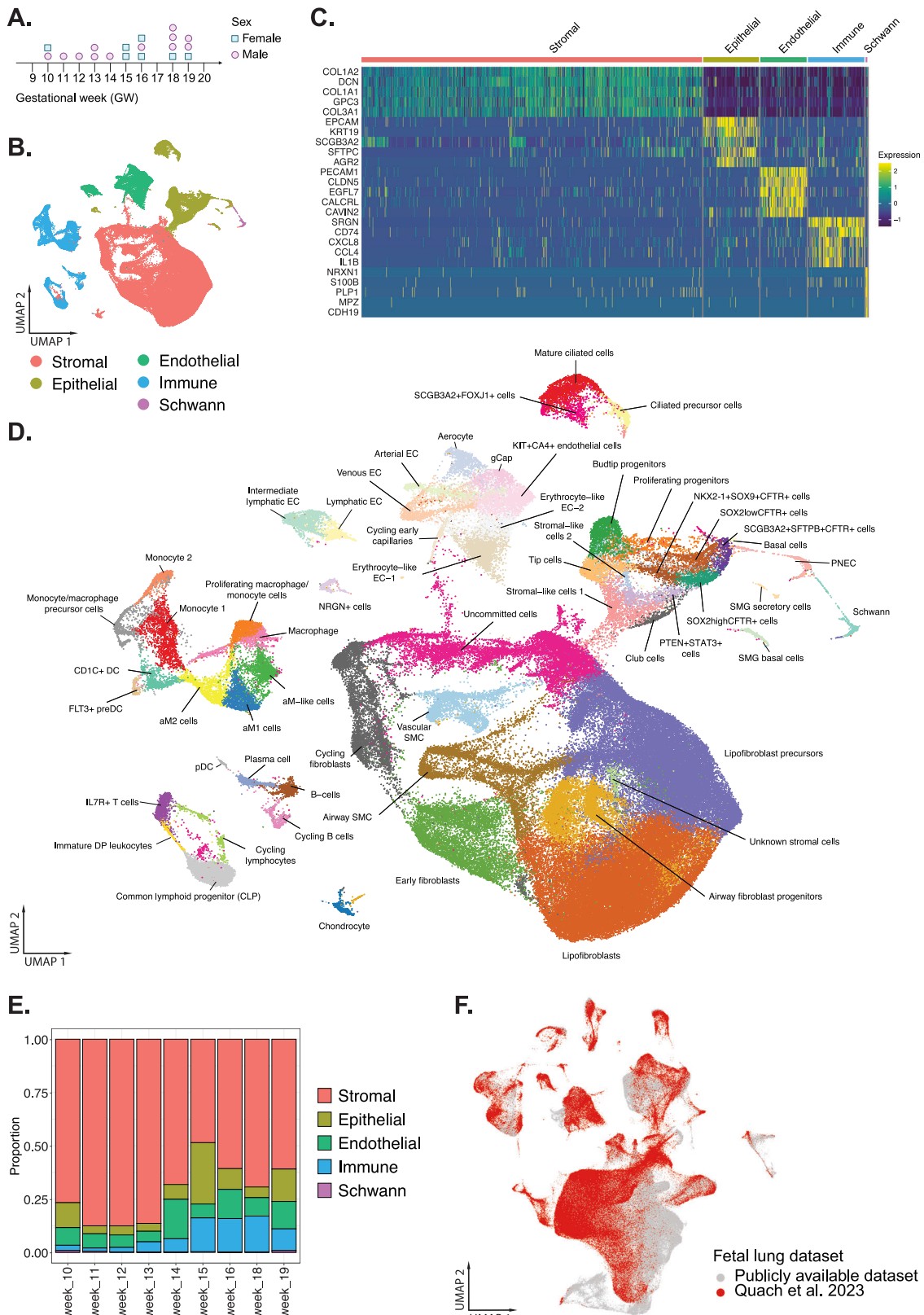

**Fig. 1 | Overview of the single-cell RNA sequencing dataset from 19 fetal lung tissues. A** Biological sex of the lungs was determined by *SRY, XIST*, and *DDX3Y* expression. The graphical image was created with Biorender.com. **B** UMAP visualization of the dataset highlighting the main cell types: stromal, epithelial, endothelial, immune, and Schwann cells. **C** Gene expression heatmap of the top 5 differentially expressed genes in each cell type. **D** UMAP projection of all 58 cell types/states identified within the integrated dataset. **E** Proportion of the major cell types across all gestational weeks. **F** Integrated UMAP projection of three published fetal lung datasets of similar developmental time points (He et al., Sountoulidis et al., and Cao et al., in gray) overlaid with our dataset (red). Source data are provided as a Source Data file.

We also used the 10X Xenium transcriptomic platform to resolve the spatial expression of a curated set of genes on archived GW15 and GW18 human fetal lung tissues. We used a custom panel consisting of 289 genes from the 10X genomics human lung gene panel and 50 custom genes informed by our single-cell RNA-sequencing dataset (Supplementary Data 1). Clustree was then used to inform Leiden clustering and cell populations (epithelial 1-2, stromal 1–5, endothelial 1, and immune 1) were determined by DEGs (Supplementary Fig. 1D–I). We identified 9 major clusters in GW15 tissue and 7 major clusters in GW18 tissue (Supplementary Fig. 1F, H). Annotated clusters were then spatially mapped to show the distinct regional distribution of each cluster (Supplementary Fig. 1J, K).

## The developing lung stroma

Stromal cells formed the largest proportion of cells in all the lung tissues analyzed (Fig. 1B). We identified 10 stromal cell subtypes (Fig. 2A) based on DEGs (Supplementary Data 1). These included lipofibroblasts, lipofibroblast precursors, cycling fibroblasts, early fibroblasts, airway fibroblast progenitors, uncommitted cells, airway smooth muscle cells (SMC), vascular SMC, chondrocytes, and an unknown stromal cell population. Lipofibroblasts and lipofibroblast precursors were the most abundant stromal cell types across all gestational weeks (Fig. 2B). Both cell types express *TCF21*, a marker previously shown in both mouse fetal and adult lung lipofibroblasts[24]. Lipofibroblast precursors differentially expressed higher levels of several canonical mesenchymal associated genes, including *PLEKHH2* and *MACF1* (Fig. 2C), similar to mouse lipofibroblasts cell lineages[25]. Regulon activity of the top differential transcription factors identified *JUN, FOS*, and *STAT3* in lipofibroblasts (Fig. 2D), which are known to regulate the synthesis of structural extracellular matrices such as collagen and elastin, and are important in alveolar development[26]. While alveolar development occurs later in gestation, it is conceivable that lipofibroblasts in early fetal lung development play a role in laying down the ECM proteins required to form airway structures. Differential regulon activity of the transcription factors driving lipofibroblast precursors include *NR2F1* and *ETV1*, which regulates cell growth, proliferation and differentiation[27,28] (Fig. 2D). Airway fibroblasts expressed high levels of *TWIST1, NFIX*, and *ZEB1*, a core epithelial-to-mesenchymal (EMT) transcription factor[12] (Fig. 2D).

We next used Xenium, high resolution multiplexed in situ spatial profiling, to determine the spatial localization of the expressed DEG in each stromal cell subtype. We identified genes associated with airway fibroblast progenitors (*SERPINF1, FBN1, IGFBP5*), lipofibroblast precursors (*MACF1* and *TCF21*), and lipofibroblasts (*EGR1, ATF3, IRF1*) in the stromal regions of the fetal lung tissues (Fig. 2E). Airway fibroblast progenitors were primarily concentrated around the large airways (epithelial 1), while lipofibroblast precursors were seen throughout the stroma. In addition, lipofibroblasts were observed scattered among the lipofibroblast precursors surrounding the epithelium.

Cycling fibroblasts and early fibroblasts were also present at a relatively high proportion in the early lung tissues (GW 10–14; ~ 6–10% and ~ 9–12% respectively) but gradually reduced in later gestational lung tissues (GW15 onwards; ~2–5% and ~3–7% respectively) (Fig. 2B and Supplementary Data 2). Other than the common collagen genes (*COL1A1, COL1A2*), cycling fibroblasts expressed high levels of genes associated with cellular proliferation *TOP2A, CENPF, MKI67* (Supplementary Data 1 and Fig. 2C). Early fibroblasts expressed high levels of genes associated with chromatin remodeling such as *HELLS* and *TYMS*[29] (Fig. 2D), previously shown to be highly expressed in mouse embryonic fibroblasts. The transcription factors with differentially high regulon activity in early fibroblasts include *E2F7, E2F8, FOXM1* (known regulators of the cell cycle in fibroblasts[30]), *TCF7*, and *E2F1*.

There was also a significant proportion of uncommitted cells that were closely associated with cycling fibroblasts and lipofibroblast precursors based on the UMAP projections (Fig. 2A). These uncommitted cells shared some transcriptional similarities to cycling fibroblasts and lipofibroblast precursors but also expressed genes associated with other cell types in general (Fig. 2C). Airway SMC differentially expressed higher levels of *TAGLN, DES*, and *MYH11*, while vascular SMC expressed *IGFBP7, MEF2C*, and *EGFL6*, both similarly reported in other studies[11]. The transcription factors and associated regulons of *TBX5, MYEF2*, and *KLF9* were differentially active in airway SMCs, while *ZFN41, PITX3*, and *HMGA1* were found in vascular SMCs. Chondrocytes differentially expressed many of the collagen genes, including *COL2A1, COL11A1, COL9A*1, and *RUNX1*, as well as *FOXC1* associated regulons known to regulate chondrocyte development[31,32]. An unknown small cluster of stromal cells (cluster 10) with distinct DEG expressed high levels of *AHSP, ALAS*, and *BPGN*, genes correlated with erythroid cells. These unknown stromal cells also expressed abundant levels of *TMEM158*, a gene enriched in cells undergoing EMT transition through STAT3 signaling[33]. Their role in the fetal lung remains undetermined.

Spatial localization of the top DEGs associated with cycling and early fibroblasts showed these cells were generally found in the stromal regions of lung tissue (Supplementary Fig. 6A). Airway SMCs were found surrounding the developing epithelium while vascular SMC genes concentrated in a distinct region presumed to be the site of a developing vessel (more evident in endothelial cluster of GW18 lung tissue). Colocalization of *THBS1* and *SOX9* was found in a cartilaginous region of the lung tissue. Immunofluorescence staining of MEF2C, DES, and ACTA2 was used to mark vascular SMC, airway SMC, and pan-SMC, respectively, and was localized to emerging blood vessels and surrounding the epithelium respectively (Supplementary Fig. 6B).

To infer developmental lineages of the fibroblast clusters, we used LatentVelo to estimate RNA velocities on a subset of the stromal cells without airway SMC, vascular SMC, and chondrocytes. A new UMAP was generated for this subset, and we visualized these velocities on the UMAP embedding and with PAGA (Fig. 3A, B). Inferred trajectories suggest that cycling fibroblasts form early fibroblasts that then contribute to airway fibroblast progenitors and lipofibroblasts. We further validated these trajectories with Slingshot, using cycling fibroblasts as the root cells or cell origins, and found a gradual change in gene expression as the cells moved along a differentiation trajectory to lipofibroblasts or airway fibroblast progenitors (Fig. 3C, D).

## A dynamically evolving fetal lung epithelium

In the fetal lung epithelium, analysis of the top DEG coupled with high expression of canonical epithelial genes associated with specific cells revealed 19 subtypes (Fig. 4A, B and Supplementary Data 1). Many of the epithelial cell types were found in all lung tissues except for ciliated cell precursors, which emerged around GW13 onwards, mature ciliated cells, and *SCGB3A2 + FOXJ1* + cells which emerged around GW14. Based on our sampling of tissues for sequencing (see "Methods"), basal cells and SMG secretory cells were not detected until around GW15, and only represented a very small fraction (~0.2–1% and ~1–5%) of the total epithelial cell population (Supplementary Data 2 and Supplementary Fig. 7A). PNEC and *SOX2*<sup>high</sup>*CFTR* + cells were found in higher abundance in early fetal lungs (up to GW13) which then decreased in proportion in later gestational tissues.

Leveraging the expression of *SOX2* and *SOX9* to determine proximal and distal airway cells respectively[34], we identified subclusters of epithelial cells that expressed high levels of *SOX2*, which included mature ciliated cells, a *CFTR*-expressing cell type that co-expressed *SCGB3A2* and *SFTPB* (*SCGB3A2 + SFTPB + CFTR +* [32]), *SCGB3A2 + FOXJ1 +* and *SOX2*<sup>high</sup>*CFTR* + progenitor cell populations (Fig. 4C). On the contrary, *SOX9* expressing cells included budtip progenitors, tip cells, *NKX2-1 + SOX9 + CFTR* + progenitor cells, and stromal-like 1 cell populations. Cells that expressed relatively lower levels of both *SOX2* and *SOX9* included SMG secretory cells, *PTEN + STAT3 +*, and *NRGN* + cells. Expression of *SOX2* in the budtip progenitors in our dataset was

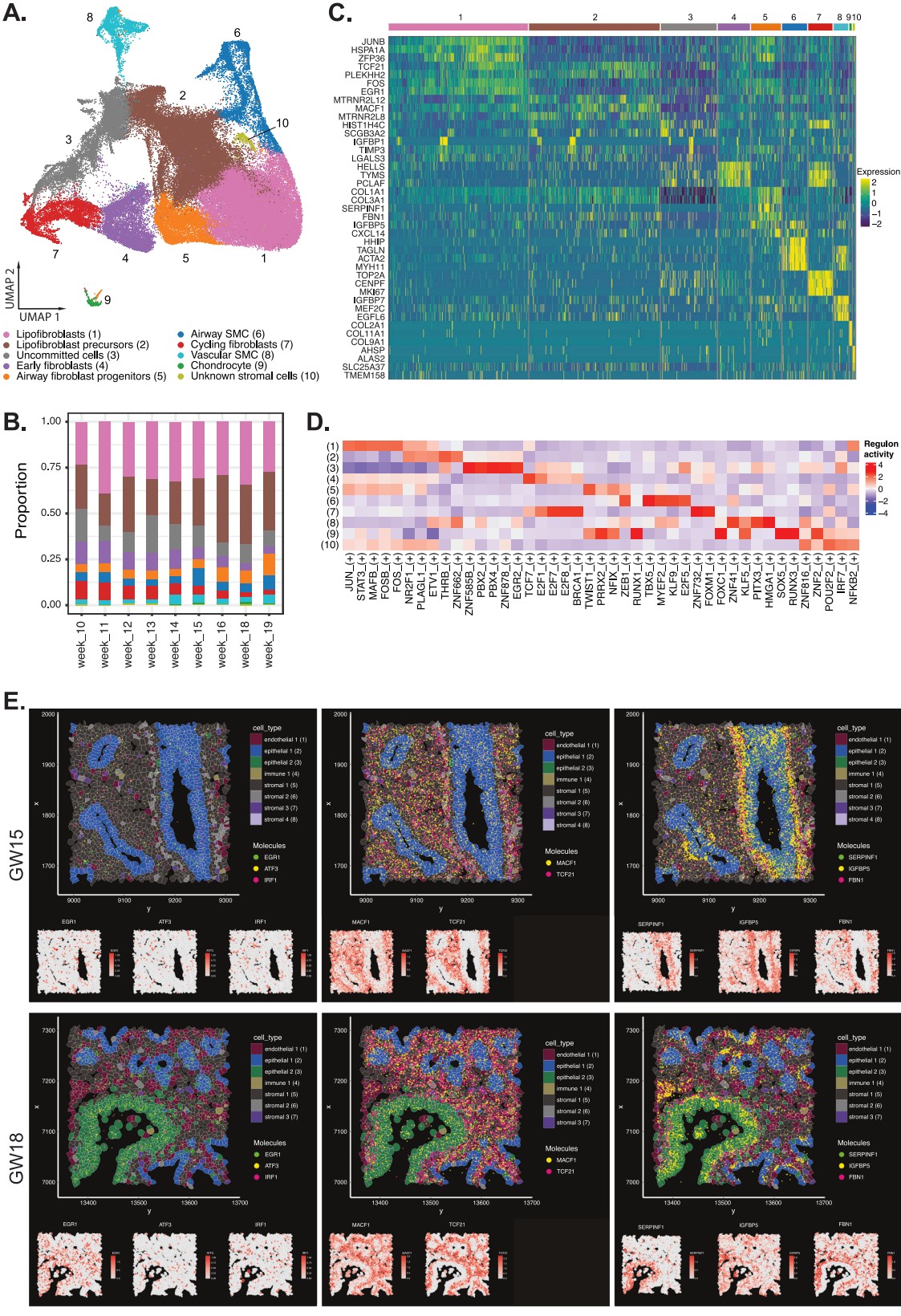

**Fig. 2 | Characterization of the stromal cell compartment identifies lipofibroblast and lipofibroblast progenitors as main cell type in the developing fetal lungs. A** UMAP visualization of the fetal stromal cell subtypes. **B** Proportion of stromal cell subtypes across gestational week. **C** Gene expression heatmap of DEGs representing each cell subtype. **D** Heatmap measuring regulon activity of the top differentially expressed transcription factor (TF) genes based on regulon specificity score (RSS) via SCENIC. **E** Xenium spatial plots of *EGR1*, *ATF3*, *IRF1* in lipofibroblasts, *TCF21* and *MACF1* in lipofibroblast precursors, and *SERPINF1*, *FBN1*, *IGFBP5* in airway fibroblast progenitors. Colors in (**B**) indicate cell type as in (**A**). Numbers in the top bar of (**C**) and (**D**) indicate cell type as in (**A**). Source data are provided as a Source Data file.

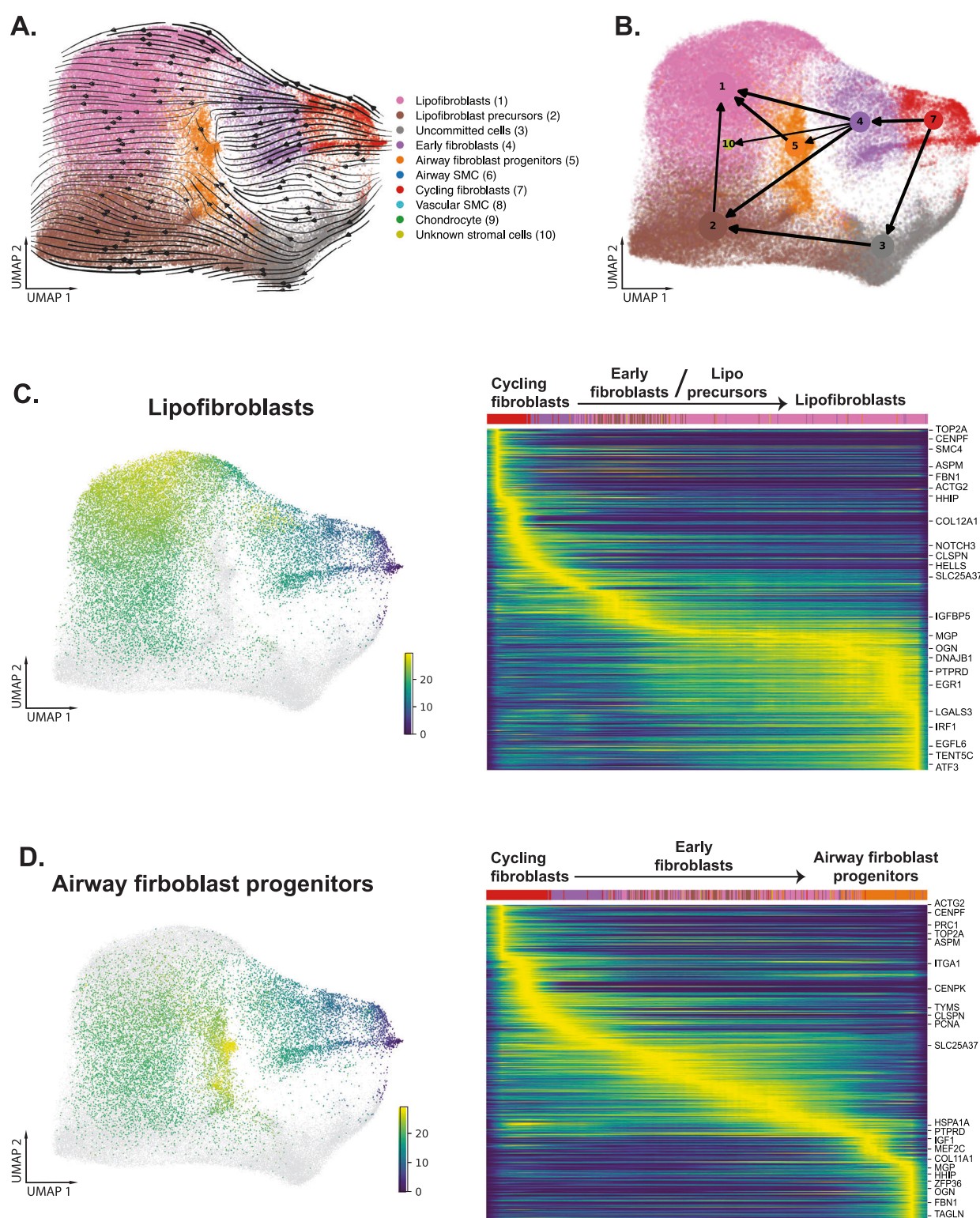

**Fig. 3 | Trajectory analysis of the stromal cell compartment show derivation of lipofibroblasts and airway fibroblast progenitors. A** LatentVelo analysis revealing inferred cell state trajectories (arrows) projected onto the UMAP. **B** Partition-based graph abstraction (PAGA) plot identified lineage trajectories informed by LatentVelo velocities. Line thickness indicates inferred transition strength. **C** Slingshot trajectory analysis with a root at cycling fibroblasts identifies a trajectory to lipofibroblasts (UMAP colored by pseudotime, dark blue to yellow). The trajectory heatmap plot shows the progression of cell types along the trajectory and the change in significantly varying genes. **D** Slingshot trajectory analysis with a root at cycling fibroblasts identifies a trajectory to airway fibroblast progenitors (UMAP colored by pseudotime, dark blue to yellow). Numbers and colors in (**B**) indicate cell type as in (**A**). Source data are provided as a Source Data file.

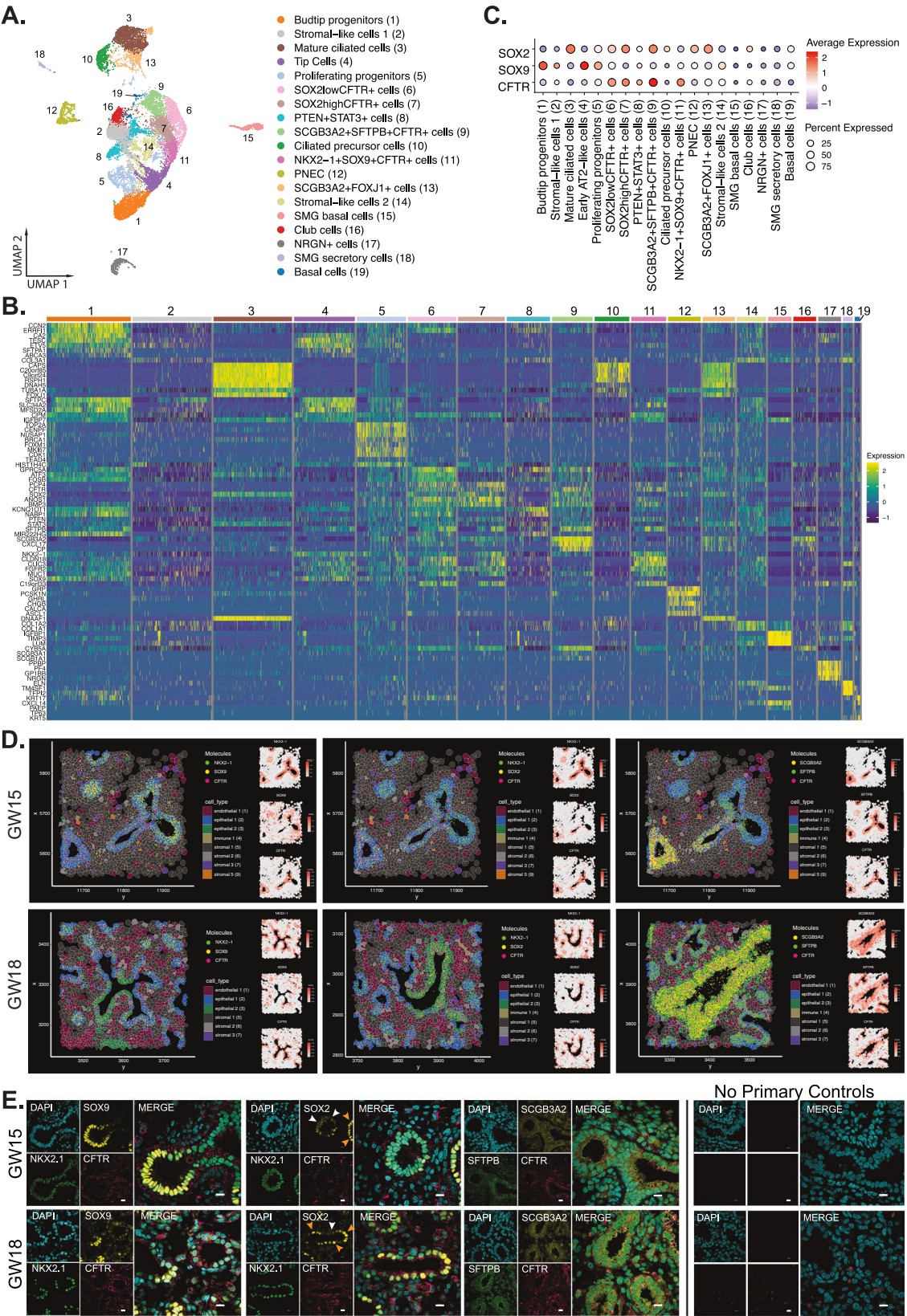

**Fig. 4 | Spatial identification of CFTR-expressing progenitor cells identified in the developing fetal lung epithelium. A** UMAP visualization of the fetal epithelial subtypes. **B** Gene expression heatmap of DEGs representing each cell subtype. **C** Dotplot of average scaled gene expression for *CFTR, SOX9*, and *SOX2* in each epithelial cell type. **D** Xenium spatial plots of *CFTR, NKX2-1, SOX9, SOX2, SCGB3A2, SFTPB* in the developing airways of GW15 (top row) and GW18 (bottom row) fetal lung tissues. **E** Immunofluorescence staining shows the localization of NKX2-1,

SOX2, SOX9, CFTR SCGB3A2, and SFTPB positive cells in GW15 (top row) and GW18 (bottom row) fetal lung tissues. At least 4 representative images were captured and analyzed for each gestational time point. The Right column panels are negative controls (no primary, secondary antibodies only). DAPI marks all nuclei. White arrowheads and orange arrowheads demarcate areas of SOX2low and SOX2high cells, respectively. Scale bar = 10 microns. Numbers in the top bar of (**B**) indicate cell type as in (**A**). Source data are provided as a Source Data file.

relatively low compared to *SOX9*, which is consistent with previous findings[35,34,36].

Proliferating progenitors were identified by the elevated expression of genes associated with cellular proliferation *MKI67* and cell cycle genes encoding the microtubule-binding protein *CENPF, NUSAP1*, and *CDK1* (Fig. 4B). These cells also expressed abundant levels of *BRCA1, MYBL2* and *FOXM1* (Supplementary Fig. 7B). Assessment of the top DEG (Supplementary Data 1) associated with these cells suggested these proliferating progenitors were relatively uncommitted.

An interesting finding in our analysis was the identification of four epithelial cell types that expressed relatively higher levels of *CFTR* (Fig. 4C, average scaled expression greater than 1). These included *NKX2-1 + SOX9 + CFTR +*, *SOX2^{high}CFTR +*, *SOX2^{low}CFTR +*, and *SCGB3A2 + SFTPB + CFTR +* cell populations. Previous studies in porcine fetal lungs suggest the role of CFTR in branching morphogenesis[37]. However, the pleiotropic roles of CFTR in the developing lung remain to be determined. The *SOX2^{low}CFTR+* shared many common DEG with *SOX2^{high}CFTR +* cells, but *SOX2^{low}CFTR +* cells also expressed *GPRC5A*, a gene involved in retinoic acid signaling instrumental to lung development[38] (Fig. 4B and Supplementary Data 1), and exhibited high regulon activity in transcription factors *NFKB1* and *IRF1* (Supplementary Fig. 7B). On the contrary, *SOX2^{high}CFTR +* cells contained elevated regulon activity in transcription factors *TFDP2* and *FOXA2*, which has been shown to mediate branching morphogenesis and differentiation[39]. Interestingly, these cells also expressed high levels of *CD47* (Supplementary Data 1), a cell surface molecule used to enrich multipotent human induced pluripotent stem cells (hiPSC)-derived lung progenitors[40].

*NKX2-1 + SOX9 + CFTR +* cells expressed high levels of *CPM, CLDN18, FGFR2*, and *MUC1*. Interestingly, *CPM* is also a cell surface molecule used to enrich fetal lung epithelial progenitors generated from hiPSC that can generate alveolospheres[41]. The association between these cells would be an interesting future study. *NKX2-1 + SOX9 + CFTR +* cells expressed high levels of the transcription factor *CEBPA*, required for lung development and activation of the alveolar development program[42], *THRA*, which activates alveolar type II cell proliferation during regeneration[43], and *SOX5*, required for branching morphogenesis in mouse lungs[44].

Spatial Xenium profiling showed *NKX2-1 + SOX9 + CFTR +* were restricted to the epithelium (Epithelial 2, blue shade) with enriched expression of all three genes in regions of bifurcations or developing bud areas of GW 15 and 18 lung tissues (Fig. 4D). In contrast, *SOX2 + CFTR +* cells were found in the stalk regions of the airways. SOX2 expression is known to regulate the cellular proliferation of epithelial cells[45], and therefore, *SOX2^{high}CFTR +* cells may be involved in the expansion of cells to form the developing airways.

Using immunofluorescence staining, we confirmed spatial expression of NKX2-1, SOX9, and CFTR triple-positive cells in the developing lung bud region (Fig. 4E). On the contrary, NKX2-1 + SOX2 + CFTR + cells were found in the fetal airways with distinct clusters of SOX2^{high} (orange arrowheads) and SOX2^{low} (white arrowheads) cells. We found no statistical significance when comparing the proportions of SOX2^{high} and SOX2^{low} cells at different time points (Supplementary Fig. 7C).

The *SCGB3A2 + SFTPB + CFTR +* cells referred to as triple positive (TP) herein, differentially expressed higher levels of *CXCL17, CP*, and *CYB5A* (Fig. 4B) and differentially high regulon activity of transcription factor *HES1*, a major NOTCH target gene, and *ASCL2*, previously shown to be a WNT/CTNNB1 transcriptional target in the gut epithelium[46] (Supplementary Fig. 7B). These TP cells were found abundantly scattered throughout the airways (Fig. 4D, E).

Unlike in adult lungs[3,47], fetal secretory, ciliated, and SMG cells expressed relatively low levels of *CFTR* compared to the progenitor cells. The rare ionocytes that express the most abundant *CFTR* transcripts in postnatal lung tissues[47] were not detected in any of the fetal

lung tissues examined here. However, this does not preclude the development of these cells later in fetal lung development. Fetal club cells expressed high levels of the canonical genes *SCGB1A1, SCGB3A2*, and *CYB5A* (Supplementary Data 1) typically found in mature club cells. On the contrary, *SCGB3A2 + FOXJ1 +* cells expressed abundant genes associated with ciliogenesis (*RSPH1, DNAH5*, Fig. 4B) suggesting these may be transitory cells originating from a secretory precursor.

Ciliated cell precursors and mature ciliated cells shared the expression of several key genes including *DNAH5*, a protein-coding gene for microtubule assembly, *FOXJ1*, a key regulator of cilia gene expression, and *RSPH1*, a gene encoding a protein that localizes cilia (Clusters 10 and 11, Fig. 4B). Both cell types also expressed high levels of regulon activity from *TP73* (Supplementary Fig. 7B), a transcriptional activator of the *FOXJ1*[48]. Ciliated cell precursors expressed high levels of *FOXN4* (Supplementary Data 1), a transcription factor required for the expression of motile cilia genes, and observed in cells undergoing ciliated cell differentiation[49]. Mature ciliated cells, on the other hand, expressed prominent levels of the *TUBA1A* required for ciliogenesis[50]. Interestingly, we found a unique cell cluster that co-expressed *SCGB3A2* and *FOXJ1* (*SCGB3A2 + FOXJ1 +*) which were found dispersed in the airways and found after GW14 (Supplementary Fig. 7D and 7E). The relationship between these cells with ciliated cells remains to be experimentally determined. However, assessment of the proximity scores of the cells from spatial transcriptomics suggests *SCGB3A2 + FOXJ1 +* cells are closely associated with other proximal cells including *SCGB3A2 + SFTPB + CFTR +, SOX2 +* cells, and club cells (low-resolution Visium Fig. 5A, high-resolution Xenium Fig. 5B), which further suggests a lineage relationship between these cells.

Tip cells and budtip progenitors both expressed genes related to surfactant protein production, such as *SFTPC, SFTPA1, ABCA3, ETV5*, and *TESC* (Fig. 4B). Budtip progenitors differentially expressed higher levels of *CCN2, ERRFL1, MIR222HG*, and genes associated with cellular proliferation *(*e.g., *FOSB)*. Budtip progenitors were found to contain differentially high regulon activity from transcription factors *NFKB1* and *SOX9* (Supplementary Fig. 7B), both involved in distal epithelial cell differentiation[51]. Transcription factors including *ELF5, CEBPA*, and *USF2* (important in regulating *SFTPA* expression[52]), were differentially active in tip cells. While *ETV5* was previously shown to regulate human distal lung identity[12] and mouse adult lung AT2 identity[53], we identified that its homolog, *ETV1*[54], was also abundantly expressed in both human fetal tip and budtip progenitor cells. Notably, we found that our tip cells and late tip cells identified by Lim et al.[55] shared similar DEGs, suggesting they may be the same or related cells. However, these cells also showed similarity to early and mid-tip cells, and therefore we chose to label them as tip cells and not define their stage explicitly. Co-localization analysis with spatial transcriptomics showed budtip progenitors and tip cells shared common genes (Fig. 4B) and were closely spatially correlated (Fig. 5A), suggesting these cells may also share regionally the same space and may represent the same cell type at different phenotypic states. Indeed, expression of *CA2* and *ETV5* was highly enriched in budtip and tip cells, while *SFTPC* was found to be enriched in budtip cells compared to neighboring tip cells (Supplementary Fig. 6C).

In our dataset, basal cells were defined by *KRT5, KRT17, TP63,* and *CXCL14* expression (Fig. 4B and Supplementary Data 1) and were rare in our dataset, especially in younger gestational age lung tissues (often found <10 cells). This may represent insufficient sampling of the larger airways, such as tracheal tissues (which was not collected for sequencing). Immunofluorescence staining for deltaNP63 (basal cell), FOXJ1 (ciliated cell), and SCGB3A2 (secretory cell) markers showed the presence of basal cells in the developing airways to be sparse in early gestational tissue but appeared more prominent in later tissues. Furthermore, *SCGB3A2 + FOXJ1 +* cells were in GW15 but not in GW12 tissues, supporting the observation in our scRNA-seq analysis that these cells emerged later. The presence of *FOXJ1* continues to increase over

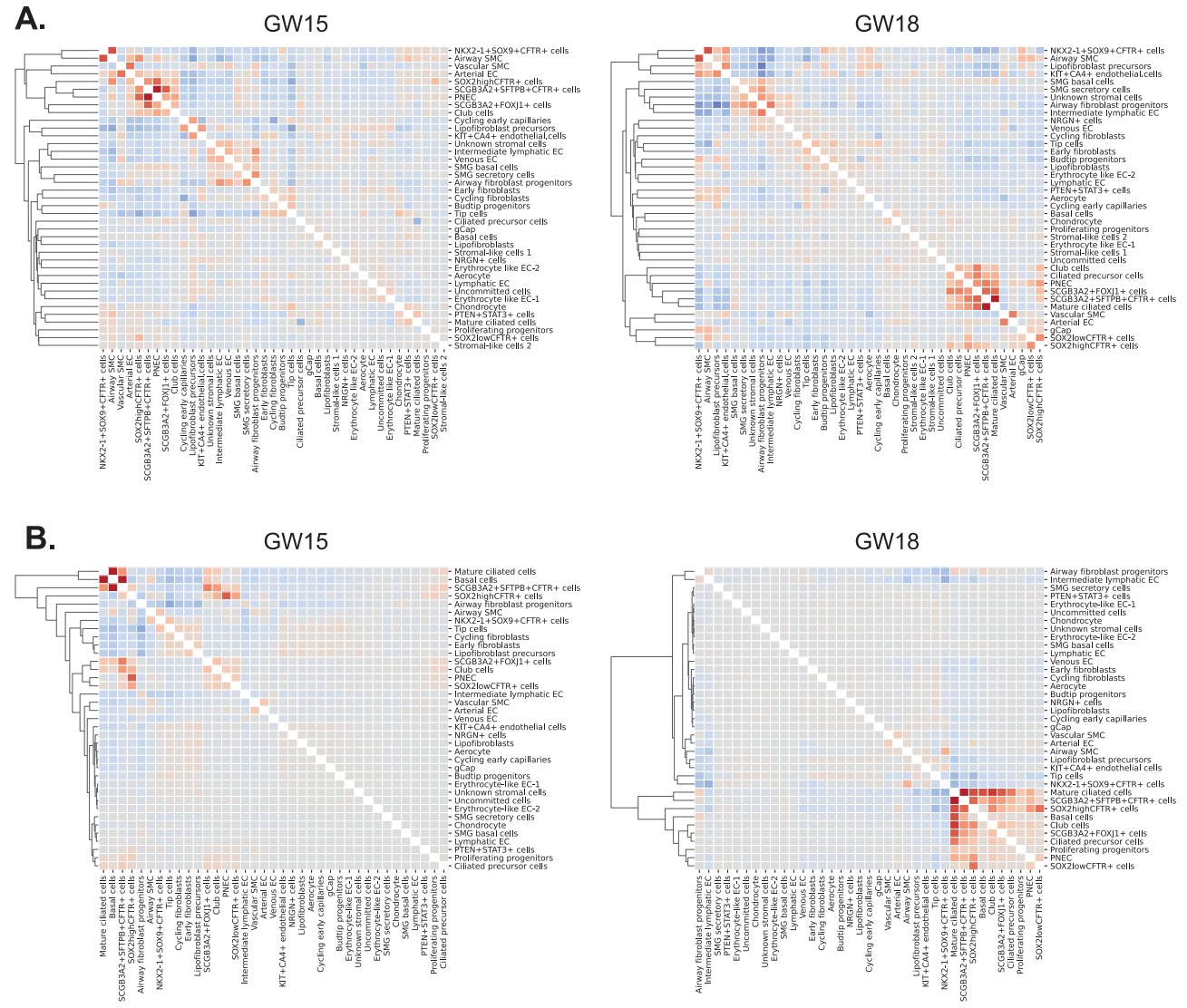

**Fig. 5 | Co-occurrence of epithelial and stromal cell types using low-resolution Visium and high-resolution Xenium spatial transcriptomics. A** Correlation between RCTD weights for Visium spots. Showing the colocalization of cell types within the same Visium spots. Hierarchical clustering shows groups of frequently colocalized cell types. **B** Xenium-informed RCTD neighborhood enrichment, showing cell types that frequently neighbor. Hierarchical clustering shows groups of frequently enriched neighbors. Source data are provided as a Source Data file.

time (Supplementary Fig. 7D). Transcription factors abundantly active in basal cells included *PITX1* and *LEF1*, both previously shown to regulate basal cell proliferation and differentiation (Supplementary Fig. 7B)[56,57].

Airway SMG basal cells expressed high levels of *IGFBP, TIMP3, LGALS1, LGALS3, S100A4,* and *S100A6,* all previously associated with SMG basal cells[10]. Similarly, SMG secretory cells expressed high levels of *UPK3B, RARRES2, C3,* and *ALDH1A2* (Supplementary Data 1). Both SMG basal and SMG secretory cells shared elevated activity of the *HOX* family transcription factors *HOXB2, HOXB4,* and *HOXA5* (Supplementary Fig. 7B), all previously shown to regulate proximal-distal patterning and epithelial differentiation in the developing mouse airways[58].

Club cells expressed high levels of *SGCB3A1, SCGB3A2, SCGB1A1,* and *CYB5A* (Fig. 4B). The *TEAD4* transcription factor was differentially active in fetal club cells (Supplementary Fig. 7B), which has previously been shown to regulate club cell development/homeostasis[59]. Club cells spatially correlated with differentiated/differentiating cells including ciliated cells, PNEC, and TP cells (Fig. 5A, B).

Similar to previous studies[10,11], PNEC emerged early in the developing lungs, but the proportion of PNEC appeared to decrease by the

end of the pseudoglandular stage ~ GW16 (Supplementary Fig. 7A). PNEC expressed high levels of the canonical genes including *CHGB, CALCA,* and *ASCL1* (Fig. 4B) and the transcription factors *NKX2-2, NEUROD1,* and *ASCL1* were most abundantly expressed in these cells (Supplementary Fig. 7B), confirming their cell identity as previously described[10,11,60]. Spatial localization for PNECs markers, *ASCL1* and *CALCA,* showed clusters of these genes in the early GW15 airways but absent in the GW18 airways (Supplementary Fig. 8A). The development of clusters of PNECs, or neuroepithelial bodies, was previously described to play a role in bombesin-mediated lung growth as these PNEC cells were found adjacent to epithelial branching points[61]. Similar to previous finding[9], two PNEC subpopulations separated by differential expression of *ASCL1* and *CALCA,* respectively, were observed in our dataset (Supplementary Fig. 8C).

An unknown epithelial cell cluster was found that expressed high levels of mTOR-related *STAT3* and *PTEN* (*PTEN + STAT3 +*). No other cell-type-specific gene was differentially expressed in these cells. In the developing mouse lungs, mTOR signaling is required for *Sox2* acquisition and conversion of *Sox9 +* distal progenitors to *Sox2 +* proximal cells[62]. It is unclear if these *PTEN + STAT3 +* cells are equivalent to

transitional cells of distal to proximal progenitors. However, these cells expressed DEG associated with cell cycle progression such as *NFKB1*, an NFKB subunit (*REL),* and *IRF1* (Supplementary Fig. 7B). Spatial transcriptomic analysis of the expression of the *PTEN, STAT3*, and *NFKB1* was used for identify these cells using Xenium (Supplementary Fig. 8B). Many stromal cells expressed all three genes but, in the epithelium, all three genes were found colocalized in the same cell and were more prominent in the stalk regions of the developing epithelium.

Other epithelial cell populations that were less defined and made up a smaller proportion of the total epithelium included the *NRGN +* cells that expressed abundant levels of *PPBP (*or *CXCL17), PF4 (*or *CXCL4),* and *ELN*, which would suggest these cells may play a role in inflammation and laying down extracellular matrices. Spatial expression of *NRGN* in GW15 fetal lungs showed a high amount of *NRGN* molecules in pockets of stromal cells surrounding the developing epithelium with few cells in the epithelium expressing *NRGN* (Supplementary Fig. 8B). In GW18 lung tissue, *NRGN* expression was most abundant in the developing endothelium with sporadic *NGRN* in some epithelial cells. It is unclear if *NRGN +* cells represent a specific epithelial cell state. Two stromal-like epithelial cell populations were also found in all fetal lung tissues examined. These cells were annotated based on the DEG of stromal associated genes, including *COL3A1*, which is expressed in squamous cell types[63]. Stromal-like cells 2 also expressed *MEOX2*[64] (Supplementary Data 1) which regulates TGFβ signaling pathway and epithelial-mesenchymal cell transition. These cells also expressed abundant *WNT2*, previously shown to orchestrate early lung morphogenesis[65]. Future studies will be needed to define the functional role of the stromal-like cells in the developing lung.

## Inferred trajectories of epithelial subtypes

Using LatentVelo, we estimated RNA velocity to predict the lineage relationships of the epithelial cell types (Fig. 6A). We also used a separate pseudotime method to arrive at these same results for validation (Supplementary Fig. 9A). We performed slingshot analysis on the late-stage tissues to determine the origins and trajectories of the differentiated cells and found trajectories from budtip progenitors through the TP cells to mature ciliated cells, PNEC, and basal/club cells (Supplementary Fig. 9B). However, subsequent validation with LatentVelo and Palantir pseudotime was unable to resolve the origin of basal cells, likely due to the low numbers of these cells in our dataset.

We then analyzed LatentVelo velocities with CellRank[66] and identified three terminal states: mature ciliated cells, PNEC, and budtip progenitor cells (Fig. 6B and Supplementary Fig. 9C, D). Using slingshot, we identified a bifurcation in the early budtip progenitor cells towards either the late budtip progenitor cells or tip cells and NKX2-1 + SOX9 + CFTR + cells (Supplementary Fig. 9E).

Based on PAGA using LatentVelo velocities, we observed trajectories that suggest a high degree of cellular plasticity and changes in cell lineage relationships (Fig. 6C). We hypothesized that epithelial lineage development may be temporally regulated. Therefore, we dissected the analysis based on developmental vignettes to unmask these dynamic events. Based on the emergence of several canonical cell types, such as mature ciliated cells and the significant decline of PNEC cells at ~ GW14 (Fig. 6D and Supplementary Fig. 4A), we grouped the tissues into 3 vignettes: Early (GW10-13), Mid (GW14–16), and Late (GW17-19). The late vignette (GW17-19) marks the early canalicular stage of lung development, a period in which differentiation and emergence of many epithelial cell subtypes emerge[67]. A few noteworthy trajectories were observed. In early gestational tissues, there were multiple sources of PNEC contributions including a trajectory from budtip progenitors through tip cells to *SOX2*^high^*CFTR* to TP and then PNEC. Changes in the DEG reflected the gradual cell fate change to PNEC differentiation (expression of *ASCL1* and *GHRL*) (Fig. 6F). Enriched GO/KEGG/Reactome terms or pathways for the cells along

the trajectory reflected the change in phenotype. During the transition from TP cells to PNEC, genes involved in FGF, NOTCH, and TGFβ signaling pathways were upregulated. All the PNEC trajectories weakened in the late stages, and the proportion of PNEC cells at these later stages decreased (Fig. 6D).

Several trajectories converged onto TP in the mid-gestational tissues (Fig. 6E). These included cell sources from *SOX2*^low^*CFTR +* cells, *SOX2*^high^*CFTR*+ cells, stromal-like cell 2 cells, proliferating progenitors, and basal cells. A strong connection (thick arrow line) between *NKX2-1 + SOX9 + CFTR +* and *SOX2*^low^*CFTR +* or *SOX2*^high^*CFTR +* cells, and the latter two with TP were observed in all time points, suggesting these cells are developmentally related. Interestingly, the connection between budtip progenitors and tip cells appeared to reflect only a subpopulation of budtip progenitor cells that emerged later in development (mid-stage). Cells that expressed abundantly higher levels of *SOX9* included budtip and tip cells (Fig. 4D). Our inferred trajectory predictions (Fig. 6F, G) and Xenium spatial localization suggest a relationship between *SOX9 +* cells. The spatial expression suggests these cells all reside close to one another in the bud or bud-adjacent region tip and next to stalk regions. We, therefore, predict that *NKX2-1 + SOX9 + CFTR +*, while they do express low levels of *SOX9*, may represent a transitional state from tip cells as they differentiate or acquire stalk identity. In assessing their DEGs, these cells are distinct from tip and *SOX2 +* cells but had similarities with both (Supplementary Fig. 9F), further supporting their lineage relationship.

During the mid (GW14–16) time points, the emergence and trajectories for mature ciliated cells were observed, stemming from budtip progenitors through tip cells, *NKX2-1 + SOX9 + CFTR +* to ciliated precursor cells or through *SOX2*^high^*CFTR +* cells, TP cells, and *SCGB3A2 + FOXJ1 +* cells to mature ciliated cells (Fig. 6G). During the transition to SCGB3A2 + FOXJ1 + cells from TP cells, genes involved in interleukin and TGFβ signaling were upregulated. Fetal club cells appeared to contribute to the ciliated cell precursor (Fig. 6E), supporting the notion that fetal club cells may have greater developmental potential as has been observed in mouse lung injury models[68]. However, this developmental trajectory was lost at later time points. Interestingly, there were no connections between fetal basal cells and club cells before GW17, which may be explained by the low numbers of basal cells captured for sequencing. At later GW, trajectories originating from basal cells contributed to PNEC and Club cell lineages (Fig. 6H), suggesting the basal cells at this stage have acquired developmental plasticity similar to fetal TP cells and adult basal cells[69].

## The plasticity of TP cells in PNEC and ciliated cell lineages

Conchola et al.[13] previously showed that TP cells can functionally contribute to PNEC and ciliated cells in the lower airways through ex vivo lineage tracing studies. In our dataset, we found a temporal contribution of these cells with a decline in PNEC in later tissues and a concomitant increase in ciliated cells around GW14 (Fig. 7A). Our trajectory analysis also suggests TP contributions to basal/club cells in later gestational tissues (>GW16). Future studies will be needed to determine the developmental potential of TP by using similar lineage tracing tools to validate these inferred trajectories[13].

CellRank was used to estimate the PNEC and mature ciliated cell fate probability for the TP cells, showing a decline in PNEC probability and increase in mature ciliated cell probability with increasing GW (Fig. 7B). Immunofluorescence staining for PNEC marker ASCL1 (yellow) and ciliated cell marker FOXJ1 (magenta) showed an abundance of PNEC cells in early GW tissue which was lost in GW18 fetal lung (Fig. 7C). Co-occurrence analysis with spatial transcriptomics showed both cell types in proximity to TP cells and *SOX2 +* cells, suggesting these cells reside in the developing stalk regions and contribute to the lineage composition of the airways (Fig. 5A, B). Together, these results demonstrate a temporal regulation of PNEC and ciliated cell

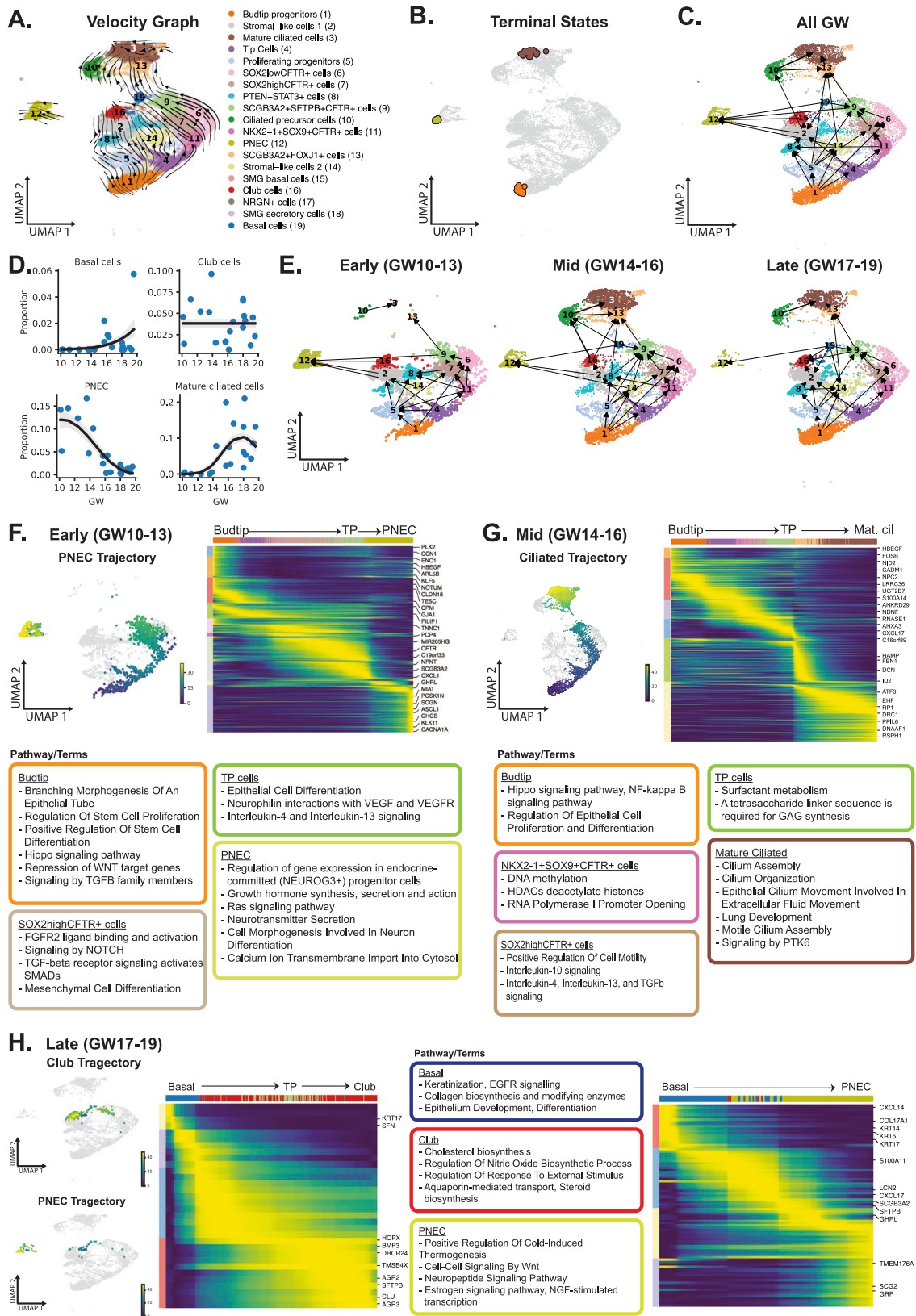

differentiation through TP cells, supporting the dynamic plasticity of these progenitor cells in the developing lung.

## Temporal changes in FGF signaling to TP

We revealed several trajectories from mid and late fetal lung tissues which suggests that cell fate acquisition is more dynamic than previously understood[70]. One explanation for these dynamic cell fate changes is that they reflect differences in the spatial microenvironment of the cell and the various signaling mechanisms involved (paracrine, autocrine, distant signaling)[71]. To determine the cellular interactions in regulating the change in gene expression and developmental potential of TP cells, we performed CellChat[72] analysis on

**Fig. 6 | RNA velocity analysis reveals dynamic epithelial cell trajectories varying across gestational ages. A** LatentVelo analysis revealing inferred cell state trajectories (arrows) projected onto the UMAP. **B** Terminal states are identified by using velocities with CellRank, shown on the UMAP. **C** Partition-based graph abstraction (PAGA) analysis reveals multiple connections with the fetal epithelia. Line thickness indicates inferred transition strength. **D** Analysis of cell type proportion over time (gestational weeks). Each point indicates a tissue sample. The best fit out of a constant, linear, or quadratic model is chosen for each cell type, shown with 95% credible intervals. **E** LatentVelo PAGA subset into Early (GW10-13), Mid (GW14–16), and Late (GW17-19). The weakening of PNEC transitions were observed, and the emergence of transitions to ciliated cells with increasing GW.

**F** Based on the PAGA, the early trajectories to PNEC from TP cells are analyzed with Slingshot (UMAP colored by pseudotime, dark blue to yellow). Trajectory heatmaps show the progression of cell types (top colored bar) and significantly varying genes along the trajectory. List of GO, KEGG, and Reactome terms and pathways enriched in each gene cluster (row colors). Pathways are shown in text boxes for the cell type that most express the genes. **G** Mid-time point Slingshot trajectories highlighting the emergence of mature ciliated cells from TP cells are shown. **H** Based on PAGA, basal cells contribute to late club and PNEC cells. Slingshot trajectories show the development of PNEC cells from basal cells and the development of Club cells from basal and TP cells. Source data are provided as a Source Data file.

our scRNA-seq dataset. Informed by the spatial relationships with spatial transcriptomics, we identified unique signaling pathways targeting the TP cells. Importantly, we observed changes in signaling when assessed through each vignette: early (GW10-13), mid (GW14–16), and late (GW18-19) (Supplementary Fig. 10A, B). Signaling pathways that specifically targeted TP cells relative to other epithelial cells, included fibroblast growth factor (FGF, Fig. 7D), WNT (Supplementary Fig. 11and Supplementary Note. 4), and NOTCH (Supplementary Fig. 12and Supplementary Note. 4) signals. FGF and NOTCH signaling pathway genes were also found to be upregulated during the transition from TP cells to PNEC with trajectory analysis (Fig. 6F).

We then focused our investigation on the FGF signaling pathway (red arrow, Supplementary Fig. 10A) as it is a known pathway involved in lung branching morphogenesis[73]. Moreover, several FGF recombinant proteins are used in directed differentiation protocols to generate fetal lung epithelia from human iPSC[16,18]. Therefore, we reasoned this pathway might play a prominent role in regulating TP cell fate. We specifically focused on cell-cell interactions through FGF signaling with TP cells (Fig. 7D). Informed by Xenium, we identified cells that were spatially in "close proximity" to TP cells which included airway fibroblast progenitors, $SOX2^{high}CFTR$ + cells, PNEC, club cells, and basal cells in GW15 lungs, as well as lipofibroblast precursors, $SCGB3A2 + FOXJ1 +$, airway SMC, and mature ciliated cells in GW18 lungs (Fig. 7E). We did not include PNEC, $SCGB3A2 + FOXJ1 +$, or mature ciliated cells (which are also spatially close to TP cells) as potential source cell types to TP, as we aimed to find the signals that may contribute to the dynamic contribution of TP cells to PNEC/ciliated cells.

We then estimated the interaction probability between specific ligand-receptor (L-R) interactions involved in FGF signaling across GWs between TP cells and FGF signal senders. We specifically looked at airway fibroblast progenitors and airway SMC as they were close in proximity with TP cells based on co-occurring cell types (Fig. 7E). We found significantly increased signaling between L-R interactions involving the ligands *FGF2* and *FGF7* expressed in airway fibroblast progenitors during in early (GW10-13) gestation, and we found significantly increased signaling for L-R interactions involving the ligand *FGF18* expressed in airway SMC during mid (GW14–16) or late gestation (GW17-19) (Fig. 7F). Gene expression of the specific FGF ligands and receptors showed elevated expression of *FGFR2* and *FGFR3* in TP cells (Supplementary Fig. 10C). We also confirmed the high expression of *FGFR2* and *FGFR3* using Visium in regions with high proportions of TP cells at GW15 and 18 (Supplementary Fig. 10D). These results suggest greater contribution of FGF2 and FGF7 signaling by airway fibroblast progenitors during early development, and FGF18 signaling by airway SMCs later in development, which may explain the temporal switch from PNEC to ciliated cells contributions.

Similar analyses of the temporal changes in WNT and NOTCH signaling can be found in Supplementary Fig. 11 and 12, respectively, with WNT signaling from airway SMCs, lipofibroblast precursors, airway fibroblast progenitors, and basal cells, and NOTCH signaling from airway SMCs, $SOX2^{high}CFTR$ + cells, and basal cells.

Overall, future studies will need to confirm the explicit role of these signaling pathways in specific cell lineage development through TP or other progenitor cell types/states and how the cell interprets the wide array of signals it receives to alter cell fate changes.

## hPSC-derived fetal lung differentiations capture cellular heterogeneity and trajectories found in the native tissue

Directed differentiation protocols of hPSC aim to capture developmental milestones ensuring robust development of bona fide cell types in cell cultures. To do this, differentiation protocols must reflect developmental processes that are observed in the primary tissues. Many differentiation protocols, including for the lung, capture "fetal-like" states, but few have benchmarked the developmental stage of these cells to fully understand the phenotype of the cells and the potential limitations of the protocols. The human "fetal" stages of tissue development span nearly 9 months of gestation, during which a small batch of early embryonic cells must proliferate, differentiate, and migrate to form a complex tissue and eventual functional organ. As the differentiations become more advanced, with the capability to generate hPSC-derived tissues-specific organoids, these models represent an exciting research tool to study fundamental mechanisms of development. This is especially important when access to primary fetal tissues for research is limited. Here, we sought to determine the developmental stage of the cells we considered to be "fetal" lung cells and organoids generated from hPSC using our most recent lung differentiation protocol[16] (Fig. 8A). We performed scRNA-seq on the hPSC-derived fetal lung cells, considered to contain mostly undifferentiated lung epithelial cells and fetal organoids, which were subjected to the next step along the differentiation pathway.

After clustering and performing DEG analysis, we compared DEGs for these hPSC cell clusters to the DEGs found in the primary fetal lung samples. In the hPSC fetal lung cells, we found significant overlap with proliferating progenitors and budtip progenitors in Clusters 0 and 1 (Fig. 8B, C). This was not surprising as the culture conditions were intended for fetal lung cell growth using retinoic acid, CHIR99021, and FGF7[36]. Other cell type DEGs that emerged in these fetal lung cell cultures included PNEC, $SOX2 +$, $SOX9 +$ progenitors, and TP cells in Cluster 2. On the contrary, hPSC-derived fetal lung organoids, which represent the next stage of differentiation, contained cells overlapping significantly with basal cells, club cells, PNEC, SMG secretory cells, ciliated cells, and TP cells (Fig. 8D, E). The high basal cell overlap in Clusters 0 and 2 was not surprising as our organoid expansion media contained dual TGFβ/SMAD signaling inhibitors: DMH1 and A83-01, which are intended to promote basal cell expansion[74].

To benchmark these hPSC differentiated cells to the gestational time point of fetal lung development, we used a Spearman correlation coefficient and found a relatively strong correlation to these early fetal lung tissues. We found that hPSC-derived fetal lung cells correlated the strongest to <GW12 lung epithelia tissue, and hPSC-derived fetal lung organoids correlated the strongest to >GW16 lung epithelia tissue (Fig. 8F). Importantly, none of the hPSC fetal lung models share a strong correlation with adult lung.

Using batch-balanced nearest neighbors, we integrated the hPSC-derived cells and the primary fetal lung epithelia (Fig. 9A). The integrated UMAP showed significant overlap of hPSC-derived fetal lung

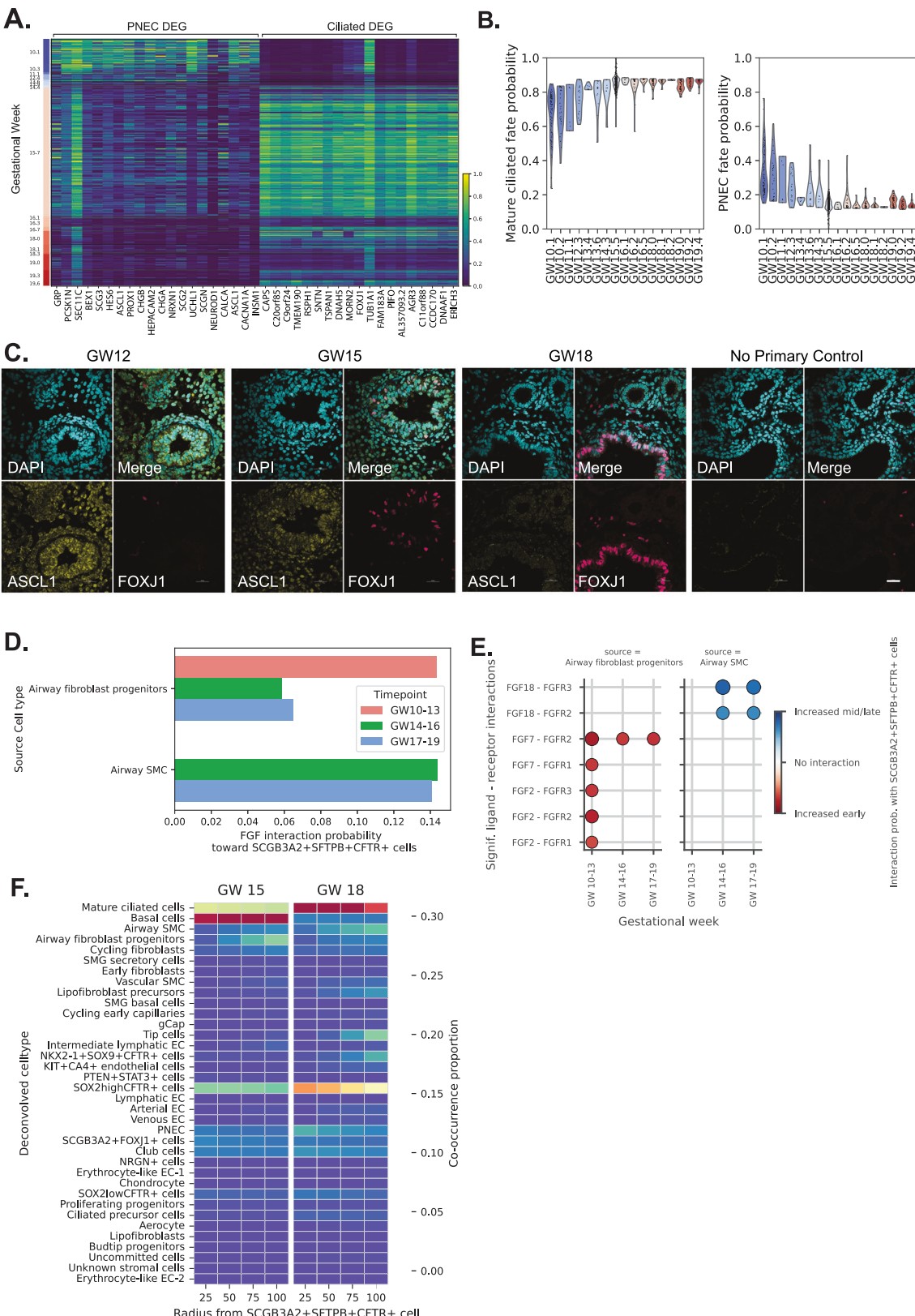

cells (purple) and organoids (green) with epithelial cells in the primary fetal lung tissue (Fig. 9B). To understand the extent to which these hPSC differentiation represent fetal lung epithelial development, we positioned these cells along our previously inferred trajectories from the native tissue. We considered the previously inferred trajectory for the primary fetal epithelium from budtip progenitors to TP cells,

through tip cells, *NKX2-1 + SOX9 + CFTR +* cells, and S*OX2^{high/low}CFTR +* cells (Fig. 9C). Using the integrated nearest neighbors graph we clustered the integrated cells and only assessed the clusters with at least 100 cells along the trajectory. Therefore, clusters with <100 cells overlapping cells between fetal and hPSC-derived cultures are not shown in the analysis (Fig. 9D).

**Fig. 7 | Regulation of TP cells differentiation and cell communication.**
**A** Heatmap showing the gene-normalized expression of the top 20 differentially expressed genes for the PNEC and mature ciliated cells within the TP cells versus gestational week. **B** CellRank matures ciliated and PNEC fate probabilities for TP cells versus gestational week (distributions shown as violin plot). **C** Immunofluorescence staining for ASCL1 (PNEC marker, yellow) and FOXJ1 (ciliated cell marker, magenta) in GW12, 15, and 18 fetal lung tissues. At least 4 representative images were captured and analyzed for each gestational time point. The Right panel shows the negative controls (no primary or secondary antibodies only). Scale = 25 microns. **D** FGF signaling interaction probabilities towards TP cells from

spatially neighboring cell types with significant FGF interactions. **E** Co-occurrence probability of deconvolved cell types within spatial neighborhoods of TP cells of a given radius with Xenium. **F** Significant ligand-receptor (L-R) interactions towards TP cells for the FGF pathway, subset to signaling cell types in proximity. Both color and point size indicate interaction probability. Red values show that L-R interactions significantly increased early. Blue values show interactions significantly increased mid/late. Significance is determined using a permutation test randomly permuting cell-type labels, significant interactions are chosen with $p < 0.01$. Source data are provided as a Source Data file.

To unbiasedly determine the epithelial cell subtype in the hPSC-derived cultures along the trajectory, we created cell type scores from the top 100 DEG for each cluster identified in the primary epithelial tissue and using the average pseudotime of each of these clusters containing hPSC fetal lung cells and organoid cells, we positioned the clusters along the trajectory (Fig. 9E, arrow above points to the direction of trajectory). These scores indicated that the fetal lung organoids became increasingly differentiated along this trajectory with the acquisition of specific genes associated with various differentiated epithelial cell types/states such as basal and TP cells. Proliferating progenitor cell type scores seen with an early pseudotime in the hPSC fetal lung cells clusters, and early hPSC organoids clusters were lost with increasing pseudotime, while $SOX2^{high/low}CFTR^+$, TP, and basal cell type scores increased in the organoids. A clear pattern of the change in gene expression emerged along the trajectory sorted by average pseudotime, with the loss of proliferating progenitor marker genes seen in the hPSC fetal lung cells (e.g., TOP2A and CENPF) and acquisition of genes identifying the more differentiated cell types such as GPC5A for $SOX2^{high/low}CFTR+$ cells, CP for TP cells, and SPRR3 for basal cells (Fig. 9F).

To further confirm this analysis of the integrated cells, we performed an independent RNA velocity analysis on the hPSC fetal lung cells and hPSC organoids using LatentVelo. Similar to the integrated trajectory analysis, we found trajectories from hPSC fetal lung cluster 0 towards clusters 1 and 2 (Fig. 9G), and trajectories from hPSC organoid clusters 1 and 4 towards clusters 0 and 2 (Fig. 9H).

Overall, our hPSC differentiated fetal cell models captured some of the epithelial cell types/states and trajectories, specifically the budtip to TP cells, along several intermediary states as observed in the primary fetal pseudoglandular/canalicular lung tissue (Fig. 9I).

In summary, we provide a high-resolution spatiotemporal atlas of the human fetal lung with a focus on the lineage relationships of the epithelial cells and interacting stromal cells in forming the developing airways (Fig. 10).

## Discussion

Reactivation of developmental mechanisms occurs during disease pathogenesis[75] (such as pulmonary fibrosis) and tissue repair[76]. Therefore, understanding the plasticity of cells under normal development and the role of the local cellular signaling environment in dictating cell fate and function may provide key insights into congenital lung diseases, chronic disease pathogenesis, and cellular responses to therapies. Here, we created a comprehensive topographic human fetal lung transcriptomic atlas of the developing human fetal lung which captured over 150,000 fetal lung cells and identified developmental trajectories that revealed remarkable cellular plasticity within the epithelial compartment. With spatially resolved transcriptomics, we identified putative temporally regulated cell signaling interactions that may dictate cell fate and behaviors within the developing lung. Finally, we showed the differentiation of hPSC progressing along similar developmental trajectories and giving rise to similar fetal cell types/states as observed in the human fetal lung epithelium. This is promising as it supports the validity of using hPSC

models for future studies in elucidating human-specific lung developmental mechanisms.

We uncovered a broader epithelial cell plasticity and with inference of lineage trajectories involving cells that express abundant CFTR. The co-expression of CFTR in these progenitor cell types suggests a putative role for CFTR in branching morphogenesis and the formation of specific epithelial cell lineages (Supplementary Note. 5). Starting from $NKX2-1 + SOX9 + CFTR +$, we found a trajectory through $SOX2^{high}CFTR +$ and $SOX2^{low}CFTR +$ cells to TP cells. We also confirmed that the TP cells have the developmental potential to generate PNEC and ciliated cells, as has previously shown with in vitro lineage tracing[13]. Moreover, our same inference model suggests TP cells may also give rise to basal and club cells later in development. Overall, our study demonstrates the multi-lineage plasticity of the CFTR-expressing population and the remarkably dynamic developmental potential of these cells. Interestingly, cellular heterogeneity and high developmental plasticity have previously been shown in endothelial lineage development[77] and neutrophil development[78]. Moreover, epithelial plasticity has also been shown to drive the formation of the endoderm during gastrulation[79]. Therefore, it is conceivable that the epithelial cells of the developing lung retained some of the high developmental plasticity of its predecessors. However, this dynamic plasticity of lung epithelial cells was not observed in the stromal compartment. Overall, we highlight how specific epithelial cell lineages are formed in the developing lungs, which may inform new hPSC-directed differentiation protocols or cell regeneration strategies for lung diseases.

Developmental processes contributing to cell fate decisions are mediated by precise cell-cell signaling interactions within a cell's microenvironment. While cells may share transcriptomic gene signatures, cell-autonomous and/or non-cell-autonomous signaling may dictate how these cells develop. Here, we focused on signaling communications between epithelial-epithelial and epithelial-stromal cells. However, it does not preclude the contributions of the immune, endothelial, and neural (Schwann) cells that are also co-developed and residing near the former two cell types. Indeed, contributions of the immune cell population to epithelial development has recently been shown in the fetal lungs[14]. Correspondingly, we saw the expression of interleukin pathways in our trajectory analysis of epithelial cells (Fig. 6), which may indicate signaling from immune cells, and supports recent findings by Barnes et al. In our cell communication and trajectory analysis, we showed temporal changes in the signaling pathways for FGF, WNT, and NOTCH targeting the TP cells. Both spatial and temporal regulation of signaling can alter the intricate microenvironment of the cell and influence its behavior and potential cell fate changes. Previous studies have demonstrated all three pathways play a role in lung branching morphogenesis and epithelial differentiation[80]. For example, lateral inhibition of NOTCH signaling increases PNEC differentiation in Hes1-null mice[81]. Similarly, we observed a dramatic decrease in PNEC in later gestational lung tissues as NOTCH signaling to TP cells increased with signaling from $SOX2^{high}CFTR +$ cells and basal cells. To confirm our lower-resolution Visium results, we used Xenium to

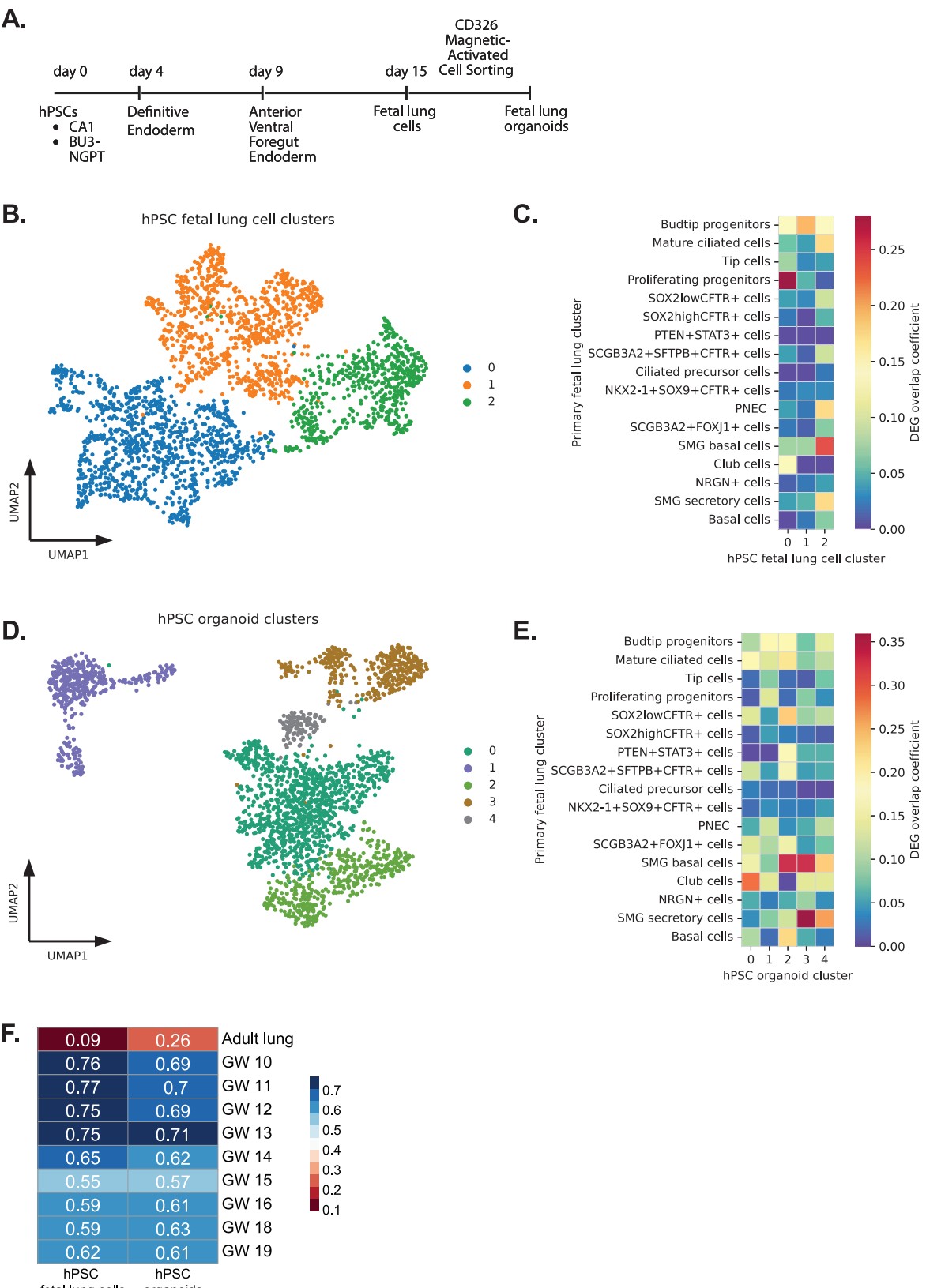

**Fig. 8 | Characterization of hPSC-derived fetal lung cells and organoids.**
**A** Schematics of stage-specific differentiation of PSC towards mature proximal airway epithelia. A graphical image was created with Biorender.com. **B** UMAP projection of hPSC-derived fetal lung cells. **C** DEG overlaps between primary fetal cell types and hPSC fetal lung clusters. **D** UMAP projection of hPSC-derived fetal lung organoids. **E** DEG overlaps between primary fetal cell types and hPSC organoid clusters. **F** Spearman's rank correlation coefficient (red to blue) analysis between hPSC-derived fetal lung models and primary-derived fetal lung epithelia and Travaglini et al. adult lung dataset using the top 50 DEG for each gestational week and adult lung. Source data are provided as a Source Data file.

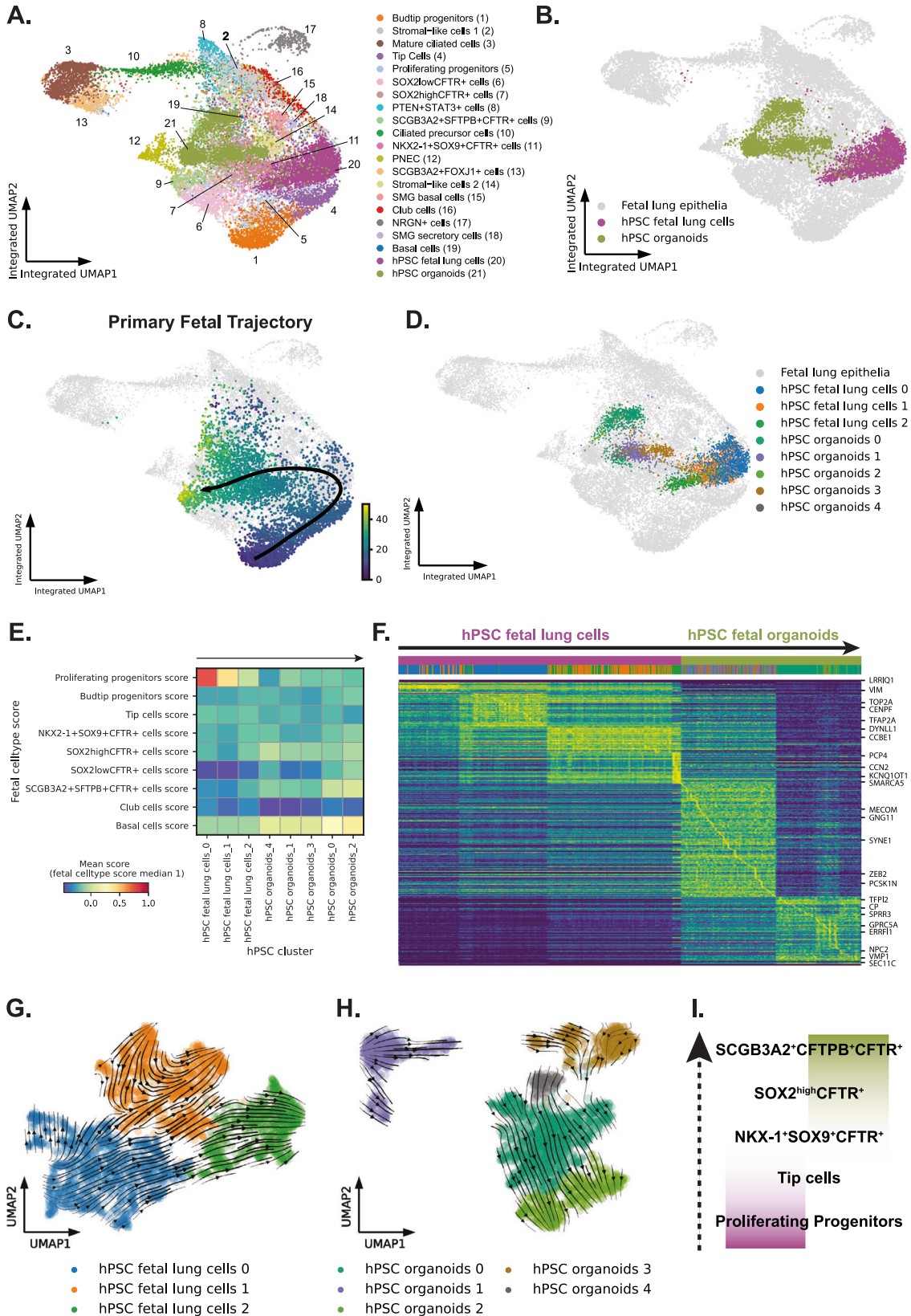

refine the spatial proximity of epithelial and stromal cell types and analyzed the putative signaling through TP.

Future studies will need to mechanistically determine the precise role of the signaling pathways in regulating specific epithelial cell lineage development possibly through cellular barcoding and single-cell lineage tracing studies. Altogether our data highlights the intricate

nature of cellular communications within a cell's microenvironment that may contribute to the heterogeneous and complex development of the fetal lung epithelium.

Our human fetal lung cell atlas does have a few limitations. First, while recent publications[10–12] have sequenced earlier fetal lungs (as early as 5PCW), our study does not capture these earlier time point

**Fig. 9 | hPSC-derived lung models recapitulate trajectories towards basal and CFTR-expressing cells from the fetal lung epithelia. A** Integrated UMAP projection of hPSC-derived fetal lung cells and organoids integrated with primary fetal lung epithelia. **B** Integrated UMAP projection of hPSC-derived fetal lung cells and organoids cells. **C** Inferred primary fetal trajectory (black line with arrow) from budtip progenitors to TP cells overlaid on the integrated UMAP embedding (UMAP colored by pseudotime, dark blue to yellow). **D** hPSC-derived clusters overlapped on integrated UMAP projection. **E** Cell type scores based on the top 100 DEGs from the primary fetal epithelia reveal high proliferating progenitor scores in the hPSC fetal lung cells and high basal scores in the hPSC organoids. Scores are normalized

so that the median score for a cell of its own type is 1. hPSC clusters from left to right increase in average pseudotime. **F** Trajectory heat map showing the change in gene expression of hPSC fetal lung cells and organoids within these clusters, ordered by increasing average pseudotime of the primary fetal cells within each cluster. **G** LatentVelo analysis revealing inferred cell state trajectories (arrows) projected onto the UMAP for hPSC fetal lung cells. **H** LatentVelo analysis revealing inferred cell state trajectories (arrows) projected onto the UMAP for hPSC organoid cells. **I** Schematics of the inferred trajectory of hPSC-derived cell types found in the fetal lung cells and organoids along the differentiation path (arrow pointing upwards). Source data are provided as a Source Data file.

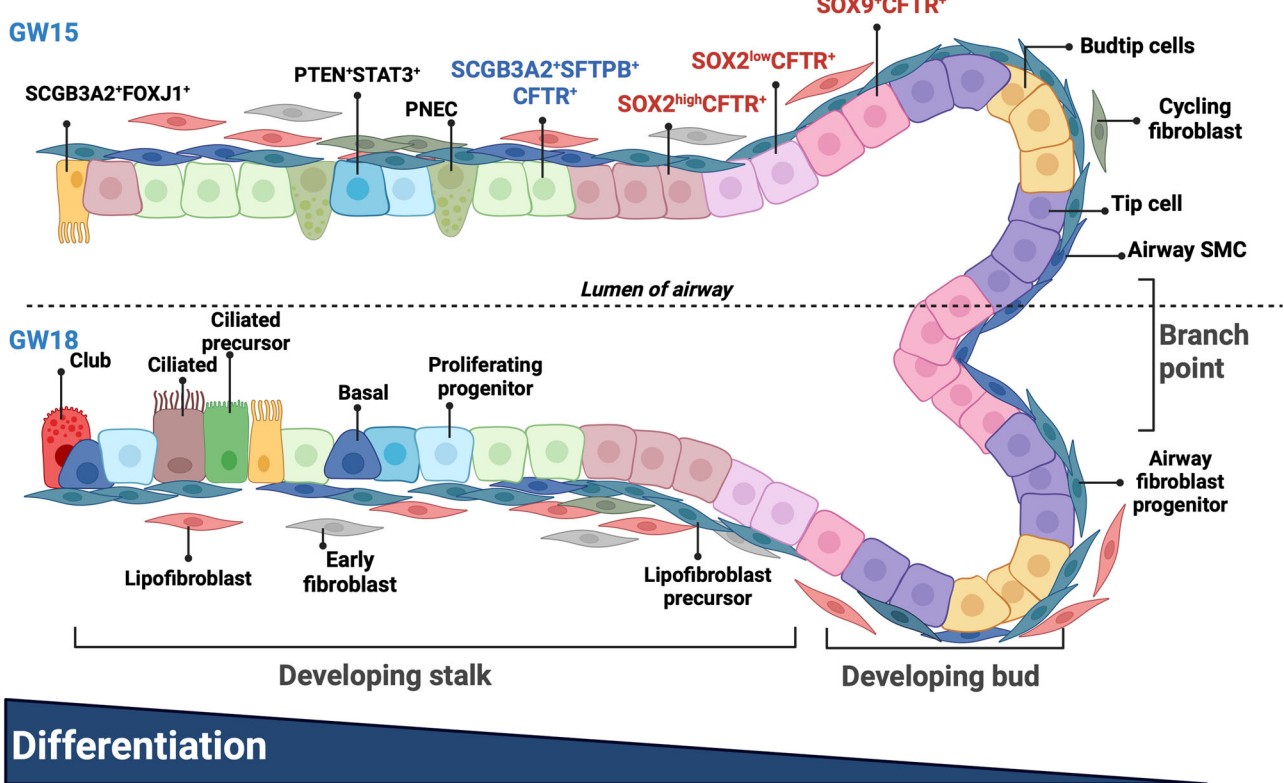

**Fig. 10 | Graphical schematic of the spatial organization of epithelial and stromal cell types in the developing fetal lung.** A schematic of the spatial localization of the epithelial and stromal cell subtypes in the developing airways illustrating the spatiotemporal changes in the developing airways. The developing distal bud region is populated by undifferentiated progenitor cell types. As the proximal stalk region develops, proximal progenitors (located in regions between the bud and stalk) generate differentiated cells that populate the developing proximal epithelium with specialized epithelial cell types. Differentiation of the developing epithelia occurs along the distal (right) to the proximal (left) axis. Cell types were identified by scRNA-seq, and their spatial localization was defined by spatial transcriptomics and immunofluorescence staining. The graphical image was created with Biorender.com.

tissues which could have uncovered additional heterogeneities. Second, our study focused on collecting and analyzing freshly isolated lung tissues in which the tracheas were not included in the sequencing analysis. The whole lungs were promptly processed for single-cell library prep with very minimal processing/transfer time, and therefore, we do not expect a significant impact on tissue processing affecting gene expression. However, since we did not include the trachea, we expect this may have skewed our analyses from identifying large airway cell types (i.e., rich basal cell source). Third, we did not perform cell enrichment for specific populations prior to library prep for the purpose of capturing less abundant cell types. However, this may have inadvertently limited our ability to detect rare cells. Moreover, due to the technical difficulties of accurately discerning "proximal" versus "distal" airways, our data could not resolve regional differences between proximal versus distal cell types, as has previously been shown[10]. Nonetheless, our data unbiasedly captured many of the cell types/states that were also found in the published fetal lung datasets,

and we further identified common and unique developmental trajectories. Finally, while our data aimed to prevent over-clustering by using Clustree to identify the optimal clustering resolution, it is possible that this method may not have resolved finer differences in cell states resulting in lower cell types/states identified. Future studies aimed at combining all of the available fetal lung datasets, much like the recently published adult lung integrated dataset[7], can provide a much more comprehensive resource and potentially capture gaps that exist in our individual datasets.

An important aspect of our study was the benchmarking of the hPSC-derived fetal lung differentiations[16]. Here, we showed that the hPSC-derived fetal lung cells and organoids captured several of the fetal-specific cell types/states and differentiation trajectories observed in the native tissue. Specifically, we showed that differentiated hPSC-derived fetal lung cells mostly represent proliferative and progenitor cells, while hPSC-derived fetal organoids contained differentiated cells representative of later stages in lung development. Since not all cell

types and trajectories were captured in the hPSC differentiation, future studies may leverage the predictions of the L-R interactions and the enrichment of specific signaling pathways to modify current differentiation protocols and improve the generation of other epithelial cell types in vitro. Overall, our work supports the use of hPSC-derived fetal lung epithelial models as a potential surrogate to study fetal lung epithelial lineage development. Importantly, hPSC differentiations are experimentally tractable models and can be used to study broader developmental ranges and fetal origins of disease, which makes benchmarking and validation of these models especially important.

In conclusion, our study identified cell types/states, trajectories, and local cellular and signaling interactomes that contribute to the dynamic development of the early human fetal lung epithelium. Understanding human fetal lung cell diversity and its role in development will improve current differentiation protocols for hPSC and thereby generate better organoid models or bona fide cell types for future therapies for congenital and chronic lung diseases.

## Methods

### Ethics statement
Human fetal lung tissue collection for the purpose of understanding human lung cell development was obtained from the Research Center for Women's and Infants' Health (RCWIH) biobank using a protocol in accordance with and approved by the Mount Sinai Hospital Research Ethics Board (REB #: 20-0035-E, March 2024) and the Hospital for Sick Children Research Ethics Board (REB #: 1000067499, January 2024). Participants did not receive compensation for their contributions. Tissue collection for research use was approved for gestational week 10–20 lung tissues. Fetal lung tissues were harvested from voluntary elective pregnancy terminations (up to 20 weeks gestation) where consent was obtained for the donation of fetal tissues for research. Only tissues from normal pregnancies were collected. No tissues collected were excluded. We collected both male and female lung tissues for our sequencing analyses. The sex of the tissue was determined retrospectively through analysis of the expression of sex-determining genes *SRY, DDX3Y,* and *XIST*. To our knowledge, there are no sex or gender effects that influence the outcome of our analysis, especially in this particularly early developmental stage of fetal development. As such, we did not perform sex-based analysis as we did not obtain more than one lung sample for some gestational weeks.

The study protocol for the use of human pluripotent stem cells (CA1 and BU3, female and male lines, respectively) for in vitro differentiation into lung cell lineages was approved by the Canadian Institutes of Health Research Stem Cell Oversight Committee and Hospital for Sick Children Research Ethics Board (REB #: 1000071246, June 2023).

### Human fetal lung collection
The gestational week was calculated based on ultrasound measurements: a combination of gestational sac diameter and crown-rump length was used to determine 10–13 weeks, whereas femur length, biparietal diameter, and foot length were used to determine >13 weeks. Freshly isolated human fetal lung tissues were collected via careful microdissection by an experienced nurse and stored in HBSS (Hank' Balanced Salt Solution) on ice during the transfer from the biobank to the Hospital for Sick Children for immediate processing. Tissues were absent of respiratory abnormalities or known genetic lung defects. A summary of the issues collected is listed in Supplementary Table 1. GW_18_1A and GW_18_1B were isolated from the upper and lower airway regions from the same tissue. Tracheal tissues were not collected for sequencing, and while we processed as much of the lung tissue as possible, sampling error may be reflected in our dataset. To estimate the post-conceptual week, subtract two weeks from the gestational week.

### Sample preparation for library construction
For all lung samples, the lung tissue, minus the trachea and a piece of the lobe, was minced and processed for single-cell isolation and RNA library preparation. The fetal lung samples were dissociated into single-cell suspension using the Multi Tissue Dissociation Kit 1 (Miltenyi Biotec; cat # 130-110-201) and 37 C Multi_tissue_dissociation B program. Dissociated cells were collected through a 40 μm pore size cell strainer (Falcon; cat # 352340) and incubated with RBC lysis buffer (Invitrogen; cat# 00-4333-57) for two minutes to remove red blood cells. Cell number and viability were assessed by trypan blue staining (Gibco; cat# 15250061) and counted with Countess II (Life Technologies, cat#A27977). This procedure resulted in cell viability of at least 80%. Approximately 10,000 cells per sample were captured and used for library construction using the 10X Chromium Next GEM single-cell 3′ Reagent Kits v3.1 (10x Genomics, cat#1000121, 1000120, 1000123). Library prep was performed as per the manufacturer's protocol. Sequencing was performed on the NovaSeq6000 (The Center for Applied Genomics (TCAG) Sequencing Facility, SickKids). The target reads were 60,000 per cell and approximately 3000–10,000 cells per sample were sequenced.

### Quality control and data processing
Raw FASTQ files were generated using supernova/cellranger mkfastq and bcl2fastq v2.20. 10x Genomics Cell Ranger v6.0.1 software was used to align reads to the human reference genome (hg 19, GRCh38). Seurat v4.0[82] was used for subsequent analysis. Cells with less than 200 features, greater than 15% mitochondrial transcript, and genes expressed in less than 3 cells were excluded from our analysis. Furthermore, principal component analysis (PCA) for each sample was done to ensure the filtering of doublets and high-quality cells. Gene expression levels were log normalized datasets and highly variable features that identify high cell-cell variation within each sample dataset were determined using Seurat's FindVariableFeatures. Subsequently features that were repeatedly variable across each sample dataset were identified as integration features. Data was then scaled (linear transformation) using Seurat's ScaleData and principal component analysis (PCA) (linear dimensional reduction) was done on each sample dataset. The generation of a master dataset was done by integrating all the sample datasets using Seurat's reciprocal PCA (rPCA) integration (3000 features). Subsequent PCA analysis was done to ensure proper integration and 30 dimensions were retained for further analysis. Further reduction was made through uniform manifold approximation and projection (UMAP) (non-linear dimensional reduction), and cluster analysis was done using Seurat's find clusters. Biological sex was determined based on *SRY, XIST*, and *DDX3Y* expression.

### Clustering and differentially expressed genes analyses
The clustree[23] package was used to inform the correct resolution for sub-clustering. Clustree was used to assess the stability of the clusters generated by taking the overlap in cell clusters across multiple resolutions and calculating the in-proportion for each edge. Resolutions with clusters derived from high in-proportion scores of multiple parent clusters were deemed over-clustered. Manual annotations were performed for each cluster and DEG compared to He et al.[10] (courtesy of Dr. Emma Rawlins)[10]. Differentially expressed genes (DEGs) for each cluster were calculated using Seurat's FindAllMarkers. Parameters that were defined included logFC > 0.25 and only output positive values. This analysis was done on the whole dataset where canonical markers identified within the top DEGs were used to assign main fetal cell type identities and within each cluster to determine DEGs in sub-clusters for sub-cluster analysis.

Cell-type scores for each cluster based on these DEGs were computed using the scanpy sc.tl.score_genes function. The top 100 DEGs for each cluster were used as the gene set, and all other genes

expressed in more than 10 cells were used as the background reference pool.

Integration analysis with publicly available datasets was done using Seurat's rPCA method to find integration anchors using default parameters, and subsequent integration using IntegrateData. Standard workflow (as above) was done to process the integrated assay and UMAP visualization.

## Query analysis

Comparisons between single-cell datasets were done and visualized using cell type label transferring and mapping UMAP cell projections via Seurat's single-cell reference mapping[76]. The processed single-cell dataset and publicly available single-cell datasets were assigned reference and query, respectively. PCA structures of the datasets were used to find anchors and classify query cells based on the highest prediction score of the reference annotations. UMAP projection was performed to visualize the cells in similar low-dimensional embeddings using Seurat's MapQuery. Similar query analysis was also done switching the reference and query dataset. Query analysis with >15% number of anchors to query cells (determined by the Seurat-associated pipeline: Azimuth[83]) were not thresholded. Analysis that did not meet the metrics underwent high confidence thresholding (>0.9).

## Gene ontology and transcription factor enrichment analysis

Gene ontology (GO) analysis was done using the 'DEenrichRPlot' function accessing EnrichR databases: GO_Biological_Process_2023, GO_Cellular_Component_2023, and GO_Molecular_Function_2023 on top 100 DEG for each sub cluster[84]. Enrichment of top 10 GO terms (ordered by log p-value). DEGs were subsequently used for transcription factor enrichment analysis using SCENIC (pySCENIC version 0.11.2)[85]. Gene regulatory interactions were calculated based on co-expression across the single-cell dataset with GRNBoost2[86], followed by pruning interactions using known TF binding motifs and the construction of dataset-specific regulatory modules (regulons)[87]. Regulons and regulon activity were then identified and scored in each individual cell using AUCell and regulon specificity score (RSS) was subsequently calculated to identify enriched transcription factors in each cluster. Pathway enrichment along the inferred epithelial trajectories was done using the REACTOME, KEGG_2021_Human, and GO_Biological_Process_2023 databases.

## Spatial transcriptomics - Visium

Spatial transcriptomics was performed using the 10X Visium platform (FFPE v2) and processed as per the manufacturer's protocol. For this, archived human fetal lung tissues (GW15 and GW18) from paraffin-embedded blocks were freshly sectioned (5 microns), and tested for RNA quality control (DV200 > 30%) before mounting on the 10X CytAssist (6.5 mm × 6.5 mm) for library prep and sequencing (~50,000 reads/spot). H&E staining was done as per the manufacturer's protocol. The Visium Human Transcriptome Probe Set v2.0 was used. Spaceranger (2.1.0) was used to perform demultiplexing and alignment to the GRCh38-2020-A reference. Spots with >10% mitochondrial content were excluded from analysis. SCTransform normalization, PCA (30 principal components were used for downstream analysis), and UMAP was done in Seurat (v4.3.0.1)[82]. Robust cell-type decomposition was done using spacxr[88]. In brief, counts and annotations from the scRNA-seq dataset were used as a reference and counts from Visium were used as a query. Spots were deconvolved using 'full' mode and RCTD weights for each spot were collected.

## Spatial transcriptomics - Xenium

For high-resolution spatial transcriptomics, 10x Xenium in situ analysis was conducted utilizing the pre-designed Xenium Human Lung Gene Expression Panel (cat #: 1000601) with an add-on custom-designed gene panel (CFVU2E, informed by the fetal lung single-cell RNA

sequencing dataset). Archived human fetal lung tissues (GW15 and GW18; same tissue used in our Visium analysis) were freshly prepared and processed according to the manufacturer's instructions. Formalin-fixed paraffin-embedded lung sections (5 microns) were mounted onto the Xenium slide. Deparaffinization and de-crosslinking were performed, and the sections were then hybridized with the custom gene probes at 50 °C overnight (~20 h), followed by post-hybridization washes. Subsequently, ligation (37 °C for 2 h) and amplification (30 °C for 2 h) were carried out using the provided ligation and amplification reagents from the manufacturer. The sections were then autofluorescence-quenched and nuclei staining was performed before loading onto the 10X Xenium Analyzer instrument for analysis. For the data analysis, raw counts were processed through the Seurat pipeline where SCTransform normalization was done prior to PCA (30 PCs) and UMAP (30 dimensions) analysis. Louvain clustering was then done at multiple resolutions, where the final resolution was decided based on clustree analysis.

Spatial co-occurrence proportions were computed by selecting all cells that were assigned by cell type deconvolution as TP cells and computing the proportion of cells in a given radius that are a particular cell type. Spatial neighbors are computed with Squidpy's[89] "sq.gr.spatial_neighbors" with the generic coordinates setting and spatial coordinates given by the centroids of the cells.

Spatial neighborhood enrichment analysis for all cell types was done with Squidpy's "sq.gr.nhood_enrichment" using a radius of 75, which uses a permutation test for the enrichment of each pair of cell types within a neighborhood. The z-score from this permutation test is reported in Fig. 5.

## Trajectory analyses

Standard RNA velocity pre-processing was done using scVelo by normalizing cells by library size with "scv.pp.filter_and_normalize", and filtering genes with less than 100 cells expressing unspliced and spliced counts for the gene with "scv.pp.filter_genes" (except in the epithelial where we use 30 and stromal where we use 300). The top 3000 highly variable genes were selected with "scv.pp.filter_genes_dispersion" with flavor = cellranger'. Following the standard scVelo preprocessing, moments were computed by averaging over 100 nearest neighbors computed on 30 principal components from the $\log(1 + x)$ transformed spliced counts with "scv.pp.moments" (300 nearest neighbors for stromal). We normalized each gene by its standard deviation and inputted it to LatentVelo for velocity inference.

We ran LatentVelo for each population separately (stromal, epithelial, endothelial, and immune populations), utilizing the batch information from each of the 19 samples. We set the latent dimension of LatentVelo as 50 and encoder hidden layer size of 75 for the endothelial and immune populations, and 100 and 125 for the epithelial and stromal. The dimension of the latent regulatory state was changed according to the complexity of the dataset and had dimensions 2 for stromal, 4 for epithelial, 3 for endothelial, and 3 for immune. When analyzing the epithelial cells with LatentVelo, we removed SMG secretory, SMG basal, and *NRGN* + cells, since these small clusters were not strongly connected to any of the larger clusters. For the stromal cells, we subset to just the fibroblast populations. A new UMAP embedding was generated for this subset.

We used standard scVelo functions to visualize velocities. To project latent velocities onto UMAP plots, velocity graphs were constructed using "scv.tl.velocity_graph" with the nearest neighbor graph constructed on the LatentVelo latent space with "scv.pp.neighbors(, use_rep = 'latent', n_neighbors = 30)", and streamlines were plotted with "scv.pl.velocity_embeding_stream". PAGA is run using latent velocities with scv.tl.paga, using the LatentVelo latent time as a prior. CellRank was used to compute terminal states. We combined the velocity kernel for LatentVelo's velocities and the connectivity kernel using the nearest neighbor graph, with weights of 0.2 and 0.8.

Slingshot[22] trajectory analysis was performed to get detailed trajectories for specific cluster subsets. Slingshot was run on 30 principal components, using the root clusters as informed by the LatentVelo analysis. To find lineage-associated genes, we used the tradeSeq[90] function associationTest, and selected genes with adjusted false discovery rate below 0.01 on each lineage separately. These genes were then used in trajectory heat maps to visualize the change in expression over pseudotime along the trajectory.

Lineage-associated genes were clustered using ward linkage of the distance matrix calculated by 1 minus the correlation matrix between the genes for cells along the trajectory. Scipy's "scipy.cluster.hierarchy.fcluster" was used to cut the hierarchical clustering tree at the specified number of clusters. Trajectory heatmaps were smoothed with a general additive model with 10 splines. We found enriched GO (GO_Biological_Process_2023 enrichr library), KEGG (KEGG_2021_Human enrichr library), and Reactome terms and pathways with adjusted p-values below 0.05.

Palantir pseudotime[91] was used to validate LatentVelo results on the epithelial subset. We chose a root at the GW10 budtip progenitors and ran Palantir with 5 components. CellRank with a pseudotime kernel was used to compute terminal states and fate probabilities.

### Ligand-receptor interaction analysis
Cellchat[72] was used to infer significant (permutation test permutating cell type labels with p-value < 0.01) ligand to receptor interactions between the fetal epithelial and stromal populations. Normalized counts and cell annotations from the scRNA-seq dataset split into early (GW10–13), mid (GW14–16), and late (GW18-19) were used as input. Source cell types were subset according to spatially close cell types to the target, as determined with Visium. Significant cell signaling pathways of interest were further analyzed. A paired Wilcoxon test for the sum of pathway interactions (information flow) was used to determine significantly varying pathways between early, mid, and late.

### Directed differentiation of hPSC to fetal lung cells and organoids
CA1 hESC (courtesy of Dr. Andras Nagy, Lunenfeld Tanenbaum Research Institute, Toronto) and BU3 hPSCs (courtesy of Dr. Darrell Kotton, Boston University) were used to generate hPSC-derived fetal lung cells and organoids using our previously established protocol[16]. In brief, cells were differentiated through a stepwise manner in STEMDiff definitive endoderm media for four days (STEMCELL technologies, cat#: 05110), anterior ventral foregut endoderm media (KnockOut DMEM (Gibco; cat#: 10829-018), 10% KnockOut serum replacement (KOSR; Gibco; cat#: 10828-028), 1% penicillin/streptomycin (Gibco; cat#: 15-140-122), 2 mM Glutamax (Gibco; cat#: 35050-061), 0.15 mM 1-mono-thioglycerol (MTG; Sigma-Aldrich; cat#: M6145), 1 mM non-essential amino acid (Gibco; cat#: 11140-050), 500 ng/ml recombinant fibroblast growth factor 2 (Peprotech; cat#: 100-18B), 50 ng/ml recombinant sonic hedgehog (Peprotech; cat#: 100-45) for five days, fetal lung epithelial progenitor media (75% IMDM (Gibco; cat#: 12440053), 25% Ham's F12 (Gibco; cat #: 11765062), 1% B-27 supplement (Invitrogen; cat#: 17504-44), 0.5% N-2 supplement (Invitrogen; cat#: 17502-048), 0.05% BSA (Invitrogen; cat#: 15260-037), 2 mM Glutamax (Gibco; cat#: 35050-061), 50 μg/ml ascorbic acid (Sigma-Aldrich; cat#: A4544), 0.15 mM 1-mono-thioglycerol (MTG; Sigma-Aldrich; cat#: M6145), 1% penicillin/streptomycin (Gibco; cat#: 15-140-122), 10 ng/ml recombinant FGF7 (KGF, Peprotech; cat#:100-19), 50 ng/ml recombinant FGF10 (Peprotech; cat#:100-26), 10 ng/ml recombinant bone morphometric protein-4 (BMP4; R&D Systems; cat#: 314-BP), 3 μM CHIR99021 (STEMCELL Technologies; cat#: 72054), and 100 nM all-trans retinoic acid (Sigma-Aldrich; cat#: R2625) for six days. To generate fetal lung organoids, fetal lung cells were enriched through magnetic activated cell sorting of CD326, embedded into matrigel spheroids, and cultured in airway expansion medium

containing 1 μM A83-01 (STEMCELL Technologies; cat#: 72022) and 1 μM DMH1 (STEMCELL Technologies; cat#: 73632) in PneumaCult-Ex Plus (STEMCELL Technologies, cat#: 05040). hPSC cells were labeled using unique MULTI-seq lipid modified oligonucleotides (Oligo 1: "GGAGAAGA"; Oligo 2: "CCACAATG"; Oligo 3: "TGAGACCT", Oligo 4: "GCACACGC") and subsequently quenched in 1% BSA in PBS[92]. Library preparation and single-cell analysis were done similarly to the above. A total of 2493 and 2450 cells of fetal lung cells and organoids were sequenced, and 23,968 and 28,648 UMIs were captured respectively. DEGs between gestational ages of the fetal lung epithelia were used to generate a panel of key features. hPSC fetal lung cells and hPSC organoids were clustered separately using sc.tl.louvain with resolutions 0.15 and 0.25, respectively. DEGs for these clusters were computed with a Wilcoxen rank-sum test and compared to the primary fetal lung clusters using sc.tl.marker_gene_overlap with the top 100 marker genes for each hPSC cluster and with the overlap coefficient (the number of shared DEG divided by the size of the smaller of the two DEG lists).

A combined embedding for the primary fetal lung epithelia and hPSC-derived cells was created with Scanpy's batch balanced nearest neighbors "sc.external.pp.bbknn"[93] and computing a UMAP embedding. Clustering on this integrated space is done with Scanpy's "sc.tl.louvain" with resolution 2.25. Only hPSC cells within integrated clusters that contained more than 100 hPSC cells were considered for trajectory analysis. Trajectory analysis was performed with a slingshot on the 30 principal components of the primary fetal cells for the subset of budtip progenitors, tip cells, $NKX2-1^+SOX9^+CFTR^+$, $SOX2^{high}CFTR^+$, and $SCGB3A2^+SFTPB^+CFTR^+$ cells. Average pseudotime was assigned to the hPSC cells within their respective integrated clusters and ordered along the trajectory.

RNA velocity was computed separately for hPSC fetal lung cells and organoids using LatentVelo with a latent dimension of 20 and a dimension of latent regulatory state of 1.

### Immunofluorescence staining
Previously archived formalin-fixed paraffin-embedded human fetal lung tissues were sectioned at 8um thickness for immunohistochemical analysis. Slides were deparaffinized using Xylene to 70% ethanol to water. Briefly, antigen retrieval was then performed using citrate buffer solution (pH 6.0) for 15 min. The slides were then briefly rinsed with 1X PBS and blocked with 6% normal donkey serum in PBS for 1 h at room temperature. In a humidified chamber, the slides were then incubated with primary antibodies (Supplementary Table 3) overnight at 4 °C and then washed 2X with PBS. Secondary antibodies were then added for 1 h at room temperature. Nuclei were counterstained with DAPI (ThermoFisher, 1:1000) for 15 min at room temperature, mounted with DAKO immunofluorescence mounting media, and kept in the dark until imaging. Fluorescence images were captured with the Nikon A1R confocal microscope and analyzed with NIS-Elements software (Nikon Instruments Inc.).

### Immunofluorescence quantification
Fluorescence images of fetal lung tissue samples at gestational weeks 12, 15, and 18 were captured at 40x magnification using the Nikon A1R confocal microscope with a Plan-Apochromat 40 × /1.25 water lens. The images were imported into Volocity software for SOX2 quantitation analysis. The 'Find Object' tool was utilized to segment cells in the channels of interest. Intensity thresholds were manually set by the user, with intensity values greater than 1081 classified as high SOX2 expressing regions. A fine filter was applied to remove noise, and 'local contrast enhancement' and 'fill holes' tools were used to aid in cell segmentation. The software calculated area measurements for high and low SOX2 populations, and the percentage of high/low SOX2 expressing areas was determined relative to the total SOX2 expressing area ($\mu m^2$).

## Statistics & reproducibility

No statistical method was used to predetermine the sample size. No data were excluded from the analyses, and the experiments were not randomized. The Investigators were not blinded to allocation during experiments and outcome assessment. For differential expression analyses of scRNA-seq datasets, the Wilcoxon ranked sum test was used, and genes were ranked based on average log fold change.

For spatial transcriptomics, we analyzed one section of a GW15 and one of a GW18 lung for both Visium and Xenium analyses.

For cell type proportion analysis, we fit binomial models with constant, linear, and quadratic dependence on gestational week for the counts of each cell type in the tissues. These models fit with the rstanarm package using a logit link function, default priors (zero-centered normal with a standard deviation of 2.5), and default MCMC settings (1000 warmup, 1000 sampling). The best model was chosen by leave-one-out cross-validation score.

For comparison of SOX2 expression, lower magnification images of SOX2 staining were acquired for GW12, 15, and 18 fetal lung tissues. Six different fields of view were acquired per gestational week (a total of 18 different FOVs). Images were imported to Volocity software to segment total SOX2, SOX2$^{high}$, and SOX2$^{low}$ expressing areas based on fluorescence intensity. A Student $t$ test was performed for each GW to determine the statistical significance of the differences in the proportion of SOX2$^{high}$ versus SOX2$^{low}$ areas on segmented immunofluorescence images.

For all immunohistochemical analyses, representative images from one lung tissue for each GW12, GW15, and GW18 were used in the figures. A minimum of four representative fields of view were captured and analyzed for each time point.

For hPSC differentiations, one human embryonic stem cell line (CA1, female) and one induced pluripotent stem cell line (BU3, male) were used. Technical replicates ($n$ = at least 3 sets of independent differentiations) were pooled for sequencing. Spearman's rank correlation coefficient of hPSC-derived fetal lung models and in vivo adult and fetal lung cells were calculated using log-normalized gene expression of the top 50 differentially expressed genes between each gestational week in our fetal lung dataset and adult lung dataset[4].

## Reporting summary

Further information on research design is available in the Nature Portfolio Reporting Summary linked to this article.

## Data availability

The single-cell RNA sequencing and spatial sequencing data generated in this study have been deposited in the NCBI GEO database under accession codes GSE264398, GSE264425, GSE264407, and GSE266789. The raw sequencing files can be accessed on the NCBI website through the accession codes and all processed sequencing data can be obtained at Synapse (https://www.synapse.org/#!Synapse:syn53437291/files/). The source data generated in this study are available at https://www.synapse.org/#!Synapse:syn59808546.

The published human fetal lung datasets were obtained from He et al.[10] (Array Express; E-MTAB-11278), Sountoulidis et al.[11] (GEO; GSE215898) and Cao et al.[12] (OMIX; OMIX003147). The mouse embryonic dataset was obtained from Negretti et al.[9] (GEO; GSE165063). The adult lung single-cell RNA sequencing dataset was obtained from the Human Cell Atlas[7] portal (https://datasets.cellxgene.cziscience.com/2aa90e63-9a6d-444d-8343-8fc2a9921797.rds). Source data are provided with this paper.

## Code availability

Code for the computational analyses is available at https://github.com/spencerfar/fetal_lung_analysis (https://doi.org/10.5281/zenodo.11231149) and https://github.com/The-HQQ/Human_fetal_lung_atlas/ (https://doi.org/10.5281/zenodo.11238128) respectively.

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

## Acknowledgements

We would like to thank Zoe (Shuk-Yee) Ngan for her help in optimizing the single-cell library prep procedures. We would also like to thank Dien Nguyen for her help in data analysis. We would also like to thank Mr. Max Nilt and Ms. Dragica Curovic from the Research Center for Women's and Infants' Health (RCWIH) BioBank for their help with the fetal lung tissue collection. We would also like to thank Mr. Alper Celik from the Center of Computation Medicine (CCM) for his guidance in the bioinformatics analyses. Computational power was enabled by support provided by Compute Canada (www.computecanada.ca) and the internal SickKids High Power Computing (HPC) platform for processing our scRNA-seq datasets. Funding support for this work was provided by the SickKids Foundation & CIHR-IHDCYH grant (NI20-1070), and the Stem Cell Network Early Career Investigator - Innovation Award (FY21/ECI23) to A.P.W. H.Q. received the University of Toronto, Data Science Institute Student Fellowship award (2022-2025) and the Ontario Graduate Studentship award (2020 and 2022). S.F. received a University of Toronto Data Sciences Institute Postdoctoral Fellowship in 2023. K.K. received the 2022 Canada Graduate Student Masters Award. Schematic illustrations for Figs. 1A, 8A, 10 and Supplementary Fig. 1A were created using BioRender.com.

## Author contributions

H.Q. and A.P.W. designed the study. H.Q. and K.K. processed the tissues. H.Q. performed the scRNA-seq experiments. H.Q. and J.L. performed the Visium and Xenium experiments while H.Q. and M.W. performed the analyses. K.K. performed the immunohistochemical staining and analyses. H.Q. and A.P.W. selected the gene panel for Xenium analysis. H.Q., S.F., M.W., X.X., P.K., A.T., and B.K. performed the computational analyses. C.E.B., F.R., S.G., and T.J.M. contributed to the conceptual design of the study and reviewed the data. H.Q., S.F., and A.P.W. wrote the manuscript. All authors read the manuscript and suggested improvements to its contents and forms. All funding support for this work was awarded to A.P.W.

## Competing interests

All authors declare no competing interests.
