## [Peer review file · Nature Communications]

Early human fetal lung atlas reveals the temporal dynamics of epithelial cell plasticityREVIEWER COMMENTS

Reviewer #1 (Remarks to the Author):

This is a well-presented manuscript with important findings for the field of lung development and potential implications for adult pathogenesis. Quach et al demonstrate that they have replicated findings from previous human fetal single-cell analysis, while adding important temporal context to cell state and fate decisions. The authors identify putative signaling pathways underlying these cell fate transitions that will be useful to the community for future experiments. Lastly, they benchmark common hPSC-derived models against their findings, highlighting the strengths and weaknesses of the models in recapitulating the fetal lung epithelium, providing a valuable resource for those seeking to strengthen cellular models. However, the manuscript could benefit from provision of additional data about the comparisons to existing datasets and cell types to provide clarity to the wider community. Additionally, characterization of single-cell findings through high resolution immunofluorescence, or RNA in situ hybridization, in tissue sections would help to strengthen many of the claims surrounding cellular localization. This manuscript will be an important resource for lung community that adds to our growing knowledge of the developing human lung.

Major comments:

1. The authors do a strong job of comparing to previous fetal single-cell atlases, as well as adult and mouse datasets. However, the use of gestational weeks (this ms), rather than post-conception weeks (pcw; Sountadlis et al and He et al), to denote sample age, as well as different names for cell types, makes a quick comparison difficult. Noting in the text that the two previously published studies reanalysed here use pcw, rather than gestational age, and explaining what the difference is (potentially up to 2 weeks of developmental age) would make understanding the similarities and differences much easier for the reader. The authors should also clarify in the methods whether they made measurements to stage the embryos/fetus collected, or used estimated gestational age.

Similarly, a table should be included in Figure 1G, or supplemental note figure 1, or as an additional table, to compare the names that the three studies give to similar/probably equivalent cells. The authors must have these data from their comparative analyses and inclusion of the cell cluster name comparison would make this a more user-friendly resource. (Note that we are NOT suggesting that the authors perform more analyses, just provide more details of the thorough analyses they have already done - if possible.)

2. Similarly, matching the cell names given by each publication to the similar cell types in this ms and in Negretti et al and Travaglini et al would assist the reader in assessing the similarities and differences that the authors found.

3. Lipofibroblasts: Tcf21 has been shown to label many mesenchymal-derived cells in the developing mouse lung PMID: 10572052. In the reference cited (ref 21), the Tcf21+ mouse cells only give rise predominantly to lipofibroblasts after E15.5, and even at that stage still contribute to other lineages. Reference 22 is the publication which most strongly suggests that TCF21 is discriminative for human lipofibroblasts (at least in the adult). It would be highly informative to localize the lipofibroblasts, and other described fibroblast cell types, via immunofluorescence or in situ hybridization to discover where they are spatially localized. Ext data 2 attempts to do this for the fibroblasts, but with the exception of the airway fibroblasts the resolution is not good enough to be particularly helpful. e.g. the developing smooth muscle is well known to be localized around the airways, but appears distributed throughout the lung in the plots. This is likely to be true for other of the cell types analysed also.

4. More broadly, while the spatial transcriptomics helps to show cellular localization, it is quite low resolution. Immunofluorescence would provide higher resolution and help to strengthen claims showing where cells localize. We suggest a few additional immunofluorescence experiments to support key claims and figures.

- In Extended Data Fig 2, to confirm the presence of specific fibroblast populations throughout the lung and concentration of airway fibroblast progenitors around the airways, immunofluorescence of both GW15 and GW18 tissue for some of these populations would provide higher resolution. (See related point above).

- The authors make a lot of the various CFTR+ populations that they have identified and their analysis is really interesting. However, some improved basic spatial analysis could

really help strengthen their manuscript. 3F does not strongly show the scattering of the SCGB3A2+SFTPB+CFTR+ cells around the airways. It also does not strongly demarcate high and low SOX2+ cells, as there seems to be intermediate expression. Additional immunofluorescence of 15 GW would help to further support the analysis, and immunofluorescence of 18 GW is needed to highlight comparisons made in the trajectory analysis in figure 4. Figure 3F could easily become a whole additional supplemental figure where low magnification views and high resolution insets were shown to allow the reader to really appreciate the spatial distribution of these cells that the authors are describing.

- Are the early AT2-like cells the “late tip” cells described in detail by Lim et al PMID: 36493780? If so, that would explain their widespread localization on the Visium data in 3D. Or are early AT2-like cells, differentiating AT2 cells by localization to the developing saccules, which retain some SOX9 mRNA? And the “late tip” cells are the TTF1+SOX9+CFTR+ population? Or are they all different cell types? Again, high resolution spatial data (or possibly more information about the comparison between cell types with earlier atlases where spatial work was done – see point 1) could clarify this.

- In Extended Data Fig. 4, Visium for 18 GW but immunofluorescence for 15 GW. This makes it difficult to compare since the resolution is different for these techniques. Immunofluorescence for neuroendocrine cells and ciliated cells at 18 GW would help to support the author’s claims regarding a switch in cell fate.

- Supp note figure 4 – it is not clear in A which are the negative controls mentioned in the legend, all panels shown seem to be positive for THBS1. Possibly all epithelial cells are? This cell type needs improved immunofluorescence given how much it is discussed in the sup note.

5. The trajectory analysis presented in figure 4 is really interesting and one of the most exciting things about the ms, but would greatly benefit from an improved spatial resolution of these different cell types that the reader is just trying to get to grips with. As it is well-established that the differentiation in lung development occurs in a proximal-distal manner, improved spatial resolution of some of these cell types could also act as strongly supporting data for the trajectory analysis. (This is just emphasizing the importance of point 4 above).

Minor comments:

1. The authors sample from 19 different lungs, noting that one lung was sampled twice to capture more regions of the lung. Where was the standard sample taken from a lung regionally located, and what did it include? This description added to the methods would be helpful.

2. Additionally, in table 1, GW_15_5_1 shows an increase from the number of sequenced cells to the filtered number of cells. Does this represent experimental error? If so, why was this sample used? 15_5 also has a large % increase of cells. Is this an error?

3. There is a strong description of the cell composition of fetal lungs, with the authors also claiming that epithelial subpopulations change over time. Table 3 reveals no discernible patterns in the changes in proportions over gestational weeks. Since the sample sizes for gestational weeks are so small ($n = 1$, $n = 2$, or $n = 3$), it should be noted that this could be due to cells lost during processing, rather than an inherent biological phenomenon. It would also be helpful to include a key in Figure 1E to show which proportions are varying – the inclusion of these cell type data may make the figure more convincing than it currently is. (What are the red, green and black boxes in Figure 1E? It is in the text, but not the figure legend.) This is a confusing sentence: Cell type proportions of these cell types showed changes in the presence of varying cell types (Supplementary Table 3).

4. The first section of the supplementary note about the comparison with the other data sets is not referred to in the main text. Although it is not clear what this adds over extended data 1 – some of the figure panels seem to reproduced between extended data 1 and supplemental note figure 1.

5. Supplementary note discussion of immune cells should refer to the new (previously BioRxiv) ms characterizing the human fetal lung immune populations PMID: 38100545. For example, this published ms also finds pDC (which is described in this ms as a newly-identified cell type in fetal lung), and there are other similarities between the analyses of the immune system here and in the recent publication.

6. We suggest similar edits to the text (as minor point 3 above) to the effect that sampling error cannot be excluded as a source of changes in cell distribution from sample-to-sample when discussing the varying proportion of fibroblasts over the fetal period in Figure 2B.

7. When describing the dynamic fetal lung epithelium, the CFTR⁺ subtypes are described as novel. This is not true. While the analyses that the authors do on this cell population are novel and very interesting, the cell population itself (SCGB3A2⁺SFTB⁺CFTR⁺) has previously been identified by Conchola et al. This is referenced later in the manuscript, but it should be referenced at the first mention of this cell type.

8. The discussion about SOX2/SOX9 co-expression in the developing human bud tips could do with mentioning that previous work relied on antibody staining, and protein and mRNA distributions can be different. Meaning that this work, and previous immunofluorescent studies, may well both be correct.

9. Fig. 3F does not denote the difference between high and low SOX2⁺ expression strongly – there should be some quantitation added.

10. In Fig. 4, TP is never denoted in the text but is used on the figure with no explanation of the abbreviation. The abbreviation should be stated with its explicit definition in the text. Additionally, in the text it is stated that basal cells were not captured in early stages of development. Is it possible Fig. 4g does not represent a late stage “acquired” developmental plasticity – the basal cells simply were not captured in the earlier timepoints?

11. In Fig. 6d, the pearson correlation coefficients are extremely similar at early stages, which would make hPSC fetal lung cells and organoids both partially representative for early stages. It should not be claimed that the organoids are a significantly worse early-stage model. However, the data do show that the hPSC organoids would be the better choice for modeling later stages of development.

Reviewer #2 (Remarks to the Author):

In the current manuscript, Quach and Farrell carry out a single-cell sequencing atlas from 19 healthy human fetal lung tissues from gestational weeks 10-19 and they identified at least 58 unique cell types/states contributing to the developing lung. Overall, the manuscript represents an interesting resource that will be well utilized in the field. The authors describe cell populations and use computational approaches to infer lineage trajectories, transcription factors and cell signaling during differentiation.

I have listed several comments below, organized into 'major' and 'minor' categories. Overall, it would be nice if the authors could test some of the hypotheses generated by the computational approaches in order to make new biological observations.

Major comments:

1. Regarding Figure 2, the authors state: "Further investigation into differentially expressed transcription factors in these cells showed uniquely higher expression of SOX2 and NRD1". This is quite surprising, given that several other studies have interrogated SOX2 in the context of human fetal lung development, both using sequencing and protein based approaches, and have not described SOX2 in mesenchyme. The authors should validate their findings using other approaches (FISH, IF) to show co-expression of SOX2 and mesenchyme specific markers, using high resolution imaging.

2. Regarding the fetal epithelium, the authors state: "Mature ciliated cells and SCGB3A2+FOXJ1+ cells emerged around GW14 whereas basal cells and SMG secretory cells emerged around GW15, but these latter cell types represented only a very small fraction (~0.2-1% and ~1-5%) of the total epithelial cell population". This statement goes

against multiple published studies, including Miller and Yu et al., *Dev Cell* (2020), He and Lim et al., *Cell* (2022), which show that basal cells are present as early as 9 PCW. Could the data in this study reflect poor representation of some regions of the lungs, for example, more proximal airways and trachea are not adequately represented in the data set shown here? The authors should carefully evaluate their claims in the context of these previously published studies.

3. The authors state: “Interestingly, while SOX2 and SOX9 co-expressing cells have previously been identified in double positive distal tip lung progenitors, the expression of SOX2 in the budtip progenitors in our dataset is relatively low compared to SOX9”. This is not surprising, and SOX9-high, SOX2-low expression (both RNA and protein) in human bud tip progenitors has been reported previously by the Al-Alam (PMID: 28971977), Rawlins (PMID: 28665271) and Spence (PMID: 29249664) laboratories. This statement should be revised so as to not sound like this finding is new and/or surprising, and prior literature should be properly cited.

4. The author claim to identify novel SOX9-low, SOX2-low cell populations, stating: “Instead, cells that appeared to express both SOX2 and SOX9 at relatively equal levels included SOX2^{low}CFTR⁺ cells, PTEN⁺STAT3⁺ cells, basal cells, and the proliferating progenitors.” However, there is no high-resolution validation of these populations. Are SOX9-low cells actually expressing protein, or is this result due to low levels of mRNA that are captured as cells transition/differentiate from the bud tip progenitor state into a new differentiated state?

5. Given the fact that CFTR expression is being used as an important defining gene for several cell populations/states, it is unclear why CFTR expression data is relegated to supplemental figure 3. Similarly, supplemental figure 3C would be very helpful for defining the cell populations in Figure 3A, given that many of the canonical, lineage defining markers are shown in Supp Fig 3C. Figure 3B and Supplemental Figure 3C should be combined into one comprehensive heatmap and/or bubble plot, which includes canonical lineage defining genes, alongside all genes used to define cells in this study. It would be helpful for gene names to be easily discernable (and not squished together as is the case in Supp Fig 3C). For example, is FOXJ1 actually ever plotted? This seems essential since the authors use FOXJ1 to define a “SCGB3A2⁺ FOXJ1⁺” cell population.

6. Based on Supplemental Figure 3B, it seems that some level of CFTR is expressed in every cell population, and the data shown contradicts the statement “A novel finding was the identification of four epithelial cell types that expressed relatively high levels of CFTR”.

7. The authors show CFTR localization using antibodies, but CFTR antibodies are notoriously unreliable. The authors should validate the specificity of this reagent and possibly confirm localization using a complimentary approach such as FISH.

8. The authors state: “Transcription factors enriched in early AT2-like cells included ETV1 and HNF1B, as previously described (ref. 52) as well as SOX9 and FOXP2”. However, Ref. 52 does not discuss ETV1 at all, and mentions HNF1, but not HNF1B. Moreover, it is unclear what the authors mean by “early AT2-like cells”, since Ref. 52 carries out “...epigenomic profiling of human AEC during differentiation from AT2 to AT1-like cells. AT2 cells were extracted from explant donor lungs that had no prior evidence of chronic lung disease and allowed to differentiate into AT1-like cells in vitro over the course of 6 days utilizing well-established protocols”. Thus, all studies in ref. 52 are carried out in adult human AT2 cells.

9. The authors state: “Several trajectories converged onto SCGB3A2+SFTPb+CFTR+ in the “mid” gestational tissues. These include cell sources from SOX2^{low}CFTR+ cells, SOX2^{high}CFTR+, stromal-like cell 2, proliferating progenitors and basal cells. A strong connection (thick arrow line) between NKX2-1+SOX9+CFTR+ and SOX2^{high}CFTR+ cells and the latter with SCGB3A2+SFTPb+CFTR+ were observed in all timepoints, suggesting these cells are developmentally related and stayed consistent through the 10 weeks.” Some experimental evidence to connect these computationally inferred trajectories would be ideal.

10. Figure 4A labels colors the clusters, and then has a key that denotes which cluster corresponds to which color. It is impossible for the reader to discern between some of the shades of similar colors. Then, Figure 4C and D use numbers to label the clusters. The numbers are far easier to identify; however a numbered key is not provided. This figure is therefore very frustrating to try and connect with the text.

11. The authors state: “Within Visium spots, spatial proximity of PNEC and SCGB3A2+SFTPb+CFTR+ cells were identified at week 15, while spatial proximity of mature ciliated cells, SCGB3A2+FOXJ1+ cells, and SCGB3A2+SFTPb+CFTR+ cells were found at week 18 (Fig. 3C and 5E).” However, at week 15, it seemed that the co-occurrence was much more common for several other cell types, including lipofibroblast precursors, which are not even of epithelial origin. This seems odd.

12. Inferred signaling networks are interesting. It would be great if the authors tested some of the hypotheses generated in the computational data. For example, does NOTCH signaling play a role in PNEC differentiation, as discussed in the text?

13. It isn't clear why iPSC-differentiated lung cells were immediately integrated with human fetal lung data? It would be much more compelling if (1) iPSC data was analyzed on its own and canonical lung markers were shown; (2) iPSC data was label transferred onto human fetal data (i.e. human fetal data is used as a search space, iPSC data is computationally projected onto the fetal data based on cell similarity).

Minor comments:

1. Figure panels are stretched and compressed in an odd manner. Compare Figure 1D and E. This is the same embedding, but stretched/compressed differently.
2. Fonts in many panels appear to be weirdly stretched. Figure 1C is a good example of this.
3. Many different fonts are used in each figure. In Figure 1, it seems that at least 6 different fonts are used in different panels.

Reviewer #3 (Remarks to the Author):

The authors present a fourth scRNAseq dataset addressing the differentiation of cell types in the embryonic human lung from w10 until gestational week 19 (w19). The previous published datasets of He et al and Sountoulidis et al, were published about a year ago and the 3rd paper (Can et al) about half a year ago. The scRNAseq data of this manuscript are of high quality and are consistent with the 2 first previous papers. The authors align their data with the 2 first previously papers and suggest some new clusters epithelial cells. in Figure 1 G. There are however more clusters published by the other papers that are not detected here Figure 1D. The previous papers claim to detect about 80 cell states. It is important that the authors include also these in figure 1 G and provide an explanation for this discrepancy in the main part of the paper. They provide a relevant short discussion in supplementary note 1 but this becomes lost in the organisation of their writing. Ideally the data of the first two papers and this one should be integrated into a single dataset and provide a new consensus clustering, which would be very useful for the human lung development field. I have not understood if the authors did that. They present a combined Umap projection in Figure 1F but it is not at all clear to me how they derived it and what kind of parameters they used for the potential integration or correlation. a more detailed description of the methods is necessary

The authors focus their analysis on the gene expression trajectories leading to fibroblast and epithelial cell differentiation. I am confused by the PAGA plot in figure 2F. Are the arrows of equal thickness indicating equal probabilities, or is it a random selection of line thickness? They look very similar. The Velo-plot in fig. 1e does not show much continuity between cluster 4 and 5. Are the PAGA plot and veloplots consistent?

In an extra round of experiments the authors use Visium spatial transcriptomics to assign their cell clusters to lung regions and to define neighbouring clusters at two developmental stages. Overall these results are consistent with previously published mapping efforts of Sountoulidis et al but lack cellular resolution. Interestingly the authors define a spatial relationship of lipofibroblasts with epithelial and endothelial cells at the tips of the branches. The authors use algorithms to predict signaling pathways among the cell types showing proximity in situ. The final question that adds unique novelty to the manuscript is the comparison of cell types in the native embryonic tissue with the cell types generated by hPSCs in vitro. The authors show integration of scRNAseq data from hPSCs at different culture time points with data from the embryonic lung and find similarities in the differentiation trajectories. The analysis would be better understood if they provide information on the numbers of in vitro grown cells, the number of detected genes and how the cells clustered by themselves. The authors make an extensive description of potential regulators in the embryonic lung and would be interesting to know if specifically these are also detected in the invitro cultured cells.

An overall suggestion for improvement is to provide specific values for the detection and fold enrichment of the DEGs listed in the supplementary table and . Also the panels showing visum data are often lacking scale bars

Reviewer #4 (Remarks to the Author):

I co-reviewed this manuscript with one of the reviewers who provided the listed reports as part of the Nature Communications initiative to facilitate training in peer review and appropriate recognition for co-reviewers.

We are very grateful to the reviewers for their constructive feedback and suggestions which we
have carefully addressed in the revised manuscript and in our response. We feel these changes
have greatly improved our paper and hope our revisions and responses are satisfactory to the
reviewers.

**Response to Reviewer #1:**

This is a well-presented manuscript with important findings for the field of lung development
and potential implications for adult pathogenesis. Quach et al demonstrate that they have
replicated findings from previous human fetal single-cell analysis, while adding important
temporal context to cell state and fate decisions. The authors identify putative signaling pathways
underlying these cell fate transitions that will be useful to the community for future experiments.
Lastly, they benchmark common hPSC-derived models against their findings, highlighting the
strengths and weaknesses of the models in recapitulating the fetal lung epithelium, providing a
valuable resource for those seeking to strengthen cellular models. However, the manuscript could
benefit from provision of additional data about the comparisons to existing datasets and cell
types to provide clarity to the wider community. Additionally, characterization of single-cell
findings through high resolution immunofluorescence, or RNA in situ hybridization, in tissue
sections would help to strengthen many of the claims surrounding cellular localization. This
manuscript will be an important resource for lung community that adds to our growing
knowledge of the developing human lung.

We thank Reviewer #1's comments and support of our resource. We have carefully included
every suggestion and addressed concerns/areas of clarification in the point-by-point responses
below.

**Major comments:**

1. "The authors do a strong job of comparing to previous fetal single-cell atlases, as well as
adult and mouse datasets. However, the use of gestational weeks (this ms), rather than post-
conception weeks (pcw; Sountadlis et al and He et al), to denote sample age, as well as different
names for cell types, makes a quick comparison difficult. Noting in the text that the two
previously published studies reanalysed here use pcw, rather than gestational age, and explaining
what the difference is (potentially up to 2 weeks of developmental age) would make
understanding the similarities and differences much easier for the reader. The authors should also
clarify in the methods whether they made measurements to stage the embryos/fetus
collected, or used estimated gestational age."

We recognize that it may be beneficial to align the sample age with published datasets. This
would require converting gestational age to post conception week (PCW). While we can perform
a 2-week subtraction of GW age to get approximate PCW age, we consulted our biobank clinic
manager and were advised that the GW annotation is most reflective of the developmental age of
the tissue. The GW was calculated based on ultrasound measurements: a combination of
gestational sac diameter and crown-rump length was used to determine 10-13 weeks whereas
femur length, biparietal diameter, and foot length was used to determine > 13 weeks. We have
included these details in the methods section (page 18, lines 709-712). We have included a line
in the methods section for readers to estimate PCW by subtracting two weeks from GW. It reads.
“To estimate post-conceptual week, simply subtract two weeks from the gestational week.” in
line 720. As such, we have decided to keep the recorded GW to retain the accuracy of the sample
age.

2. “Similarly, a table should be included in Figure 1G, or supplemental note figure 1, or as
an additional table, to compare the names that the three studies give to similar/probably
equivalent cells. The authors must have these data from their comparative analyses and inclusion
of the cell cluster name comparison would make this a more user-friendly resource. (Note that
we are NOT suggesting that the authors perform more analyses, just provide more details of the
thorough analyses they have already done - if possible.) Similarly, matching the cell names given
by each publication to the similar cell types in this ms and in Negretti et al and Travaglini et al
would assist the reader in assessing the similarities and differences that the authors found.”

We appreciate the reviewer’s comments and understand the need for comparing nomenclature of
all available resources to make it more user-friendly. First to address this, we integrated our
dataset to the combined datasets by Peng He et al (ref 10), Cao et al (ref 12), and Sountoulidis et
al (ref 11) to show the overlaps between our dataset with these three atlases of similar
developmental timepoints (new **Figure 1F**). We used this to identify cells not captured in the
combined three dataset, understanding that this may reflect differences in cluster resolution and
annotations. We then performed reverse query analyses (new **Supplementary Note 1**) using the
Seurat MapQuery function in which we map our dataset (query dataset) onto the published
datasets by He, Cao and Sountoulidis (reference datasets). Here we show that our dataset does
capture many of the cell types reported in He and Sountoulidis et al but less so with Cao et al.
This may be explained by the fact that Cao et al collected and analyzed earlier fetal lung tissues
(4-8 PCW) that we were not able to collect. Therefore, some of the differentially expressed genes
(DEG) associated with these early cell types/states are either not significantly expressed or the
cells do not exist in our dataset. We have described these additional analyses in the manuscript
and detailed these additional analyses in the methods section (Page 19-20, lines 777-787).

Moreover, we have performed the same Query analyses comparing our dataset to the mouse
embryonic lung from Negretti et al new **Supplementary Fig. 2C** and now included the query
analysis comparing our dataset to adult lung tissues from a new, larger dataset from the HCA
(PMID: 37291214) in new **Supplementary Fig. 2A and B**.

Future efforts will include integrating all the fetal lung single cell datasets and establish a
consensus for determining the optimal resolution to define cell cluster numbers and cell cluster
annotations. We have chosen to use an unbiased approach informed by Clustree analysis to
determine a cluster resolution that contains stable and distinct clusters and avoids unstable
clusters caused by overclustering.

However, in integrating all the datasets, this would be beyond the scope of this study, but we
plan to work with HCA who has the computational power and experience to perform these types
of analyses in combining datasets from different groups.

3. “Lipofibroblasts: Tcf21 has been shown to label many mesenchymal-derived cells in the
developing mouse lung PMID: 10572052. In the reference cited (ref 21), the Tcf21+ mouse cells
only give rise predominantly to lipofibroblasts after E15.5, and even at that stage still contribute
to other lineages. Reference 22 is the publication which most strongly suggests that TCF21 is
discriminative for human lipofibroblasts (at least in the adult). It would be highly informative to
localize the lipofibroblasts, and other described fibroblast cell types, via immunofluorescence or
in situ hybridization to discover where they are spatially localized. Ext data 2 attempts to do this
for the fibroblasts, but with the exception of the airway fibroblasts the resolution is not good
enough to be particularly helpful. e.g. the developing smooth muscle is well known to be
localized around the airways, but appears distributed throughout the lung in the plots. This is
likely to be true for other of the cell types analyzed also.”

We have included 10X Xenium high-resolution spatial resolution images of the genes
expressed in the lipofibroblast population for the following genes (*TCF21* and *MACF1*) to show
the spatial localization of these genes in the lung tissue (new **Figure 2E**). We also show spatial
expression of the top DEGs associated with additional stromal cell types in new **Supplementary**
**fig. 3A** and performed immunofluorescence staining to confirm protein expression of genes
demarcating airway versus vascular SMC in new **Supplementary fig. 3B**. Descriptions of the
spatial expression of these genes associated with their respective cell types are now updated in
the main text.

4. “More broadly, while the spatial transcriptomics helps to show cellular localization, it is
quite low resolution. Immunofluorescence would provide higher resolution and help to

strengthen claims showing where cells localize. We suggest a few additional
immunofluorescence experiments to support key claims and figures.

We have performed additional immunofluorescence staining experiments to support the
following:

1. FOXP1 (ciliated cells) and ASCL1 (PNEC) for GW12, 15 and 18 support claims
regarding the PNEC's cell fate switch to ciliated cells (**Figure 7C**)

2. MEF2C (vascular SMC), DES (airway SMC) and ACTA2 (pan-SMC) for GW 12, 15
and 18 demonstrate the distinction in location between the SMC types (i.e., blood
vessels vs. airways) in our fetal tissue samples (**Supplementary Fig. 3C**)

3. SCGB3A2 (secretory cells), FOXP1(ciliated cells) and dNP63 (basal cell) for GW 12,
15 and 18 highlights ciliated cell differentiation over time. Specifically, the
emergence of SCGB3A2+FOXP1+ cells after GW14. Further basal cell staining
demonstrates the low proportion of basal cells observed in the small airways, with
stronger staining seen in GW15 (**Supplementary Fig. 4D**).

New high-resolution spatial expressions of the top DEGs are now provided in **Figure 2E** and
**Supplementary Fig. 3A**.

In Extended Data Fig 2, to confirm the presence of specific fibroblast populations throughout the
lung and concentration of airway fibroblast progenitors around the airways, immunofluorescence
of both GW15 and GW18 tissue for some of these populations would provide higher resolution.
(See related point above).”

Original low resolution Visium images have now been replaced with high resolution
Xenium images for both GW15 and GW18 fetal lung tissue as suggested.

5. “The authors make a lot of the various CFTR+ populations that they have identified and their
analysis is really interesting. However, some improved basic spatial analysis could really help
strengthen their manuscript. 3F does not strongly show the scattering of the
SCGB3A2+SFTPB+CFTR+ cells around the airways. It also does not strongly demarcate high
and low SOX2+ cells, as there seems to be intermediate expression. Additional
immunofluorescence of 15 GW would help to further support the analysis, and
immunofluorescence of 18 GW is needed to highlight comparisons made in the trajectory
analysis in figure 4. Figure 3F could easily become a whole additional supplemental figure
where low magnification views and high resolution insets were shown to allow the reader to
really appreciate the spatial distribution of these cells that the authors are describing.”

We have included high resolution spatial expression using the Xenium platform to show the
spatial localization of the genes associated with the various epithelial subtypes including the
*SCGB3A2+SFTPB+CFTR+* population in new Figure 4D from GW15 and GW18 lung tissue to
compliment the immunostaining data in new **Figure 4E**. The Xenium images show high
resolution expression of each gene (*SCGB3A2*, *SFTPB* and *CFTR*) in the same cell
predominantly in the developing stalk region of the airway epithelium.

Additionally, we performed quantification of SOX2^{high} and SOX2^{low} expression using IF
(**Supplementary fig. 4F**) Lower magnification images of SOX2 staining were acquired for
GW12, 15 and 18 fetal lung tissues. Six different fields of views were acquired per gestational
155 week (total of 18 different FOVs). Images were imported to Velocity software to segment total
156 SOX2, SOX2^{high} and SOX2^{low} expressing areas based on fluorescence intensity. This
quantification method is now described in the Methods section on page 24, lines 940-949. Using
this approach, we identified the presence of both high and low SOX2-expressing cells across
gestational weeks and within airways. However, there is no statistically significant difference
based on student's t-test between the proportion of high and low SOX2 within each GW.

“Are the early AT2-like cells the “late tip” cells described in detail by Lim et al PMID:
36493780? If so, that would explain their widespread localization on the Visium data in 3D. Or
are early AT2-like cells, differentiating AT2 cells by localization to the developing saccules,
which retain some SOX9 mRNA? And the “late tip” cells are the TTF1+SOX9+CFTR+
population? Or are they all different cell types? Again, high resolution spatial data (or possibly
more information about the comparison between cell types with earlier atlases where spatial
work was done – see point 1) could clarify this.”

We thank the reviewer for this suggestion and performed a direct comparison of the “early AT2-
like” and budtip progenitor cells with the reported “late tip” cells by Lim et al (see figure below)
and found that indeed both of these cell types share similar transcriptomics to the described “late
tip” cells. While these cells show the most similarity to “late-tip”, they also show similarity to
early and mid-tip (see figure below), and therefore we have decided to label the “Early AT2-
like” cells as “Tip” cells and not define their stage explicitly. We have included this on page 9,
lines 347-364.

In addition, the “late tip” cells are not identical to the *NKX2-1+SOX9+CFTR+* cells. Instead,
these *NKX2-1+SOX9+CFTR+* cells are most similar to the AT1, AT2, and late stalk cells from
He et al.

“In Extended Data Fig. 4, Visium for 18 GW but immunofluorescence for 15 GW. This makes it
 difficult to compare since the resolution is different for these techniques. Immunofluorescence
 for neuroendocrine cells and ciliated cells at 18 GW would help to support the author’s claims
 regarding a switch in cell fate.”

We had added additional immunofluorescence staining for neuroendocrine (MASH1) and
 ciliated cells (FOXJ1) in gestational weeks 12, 15 and 18 to show the temporal emergence of
 ciliated cells and the gradual “disappearance” of PNEC. These stainings are now added to new
 **Figure 7C**.

“Supp note figure 4 – it is not clear in A which are the negative controls mentioned in the legend,
 all panels shown seem to be positive for THBS1. Possibly all epithelial cells are? This cell type
 needs improved immunofluorescence given how much it is discussed in the sup note.”

THBS1 was used to delineate *PTEN*+*STAT3*+ cells based on it being one of the top DEG.
 However, we agree that the antibody is likely not specific and appears to broadly stain all cells.
 As such we have chosen to disregard the antibody staining characterization of THBS1 in our
 revision and focused on spatial transcriptomics to determine the spatial colocalization of *PTEN*,
 *STAT3* and *NFKB1* in the epithelium (new **Supplementary fig. 5B**).

6. “The trajectory analysis presented in figure 4 is really interesting and one of the most
 exciting things about the ms, but would greatly benefit from an improved spatial resolution of
 these different cell types that the reader is just trying to get to grips with. As it is well-established
 that the differentiation in lung development occurs in a proximal-distal manner, improved spatial
 resolution of some of these cell types could also act as strongly supporting data for the trajectory
 analysis. (This is just emphasizing the importance of point 4 above).”

We completely agree with this comment and have included high resolution spatial
 transcriptomics using the 10X Xenium platform to show the spatial localization of the different
 cell types in the developing airways (new **Figure 4D**, new **Figure 7E**, Extended **Figure 5A**). We
 have also used the Xenium data to inform co-occurrence of cell types (cells that are spatially
 close to one another) in new **Figure 5B and 7E** and showed mature ciliated cells,
 *SCGB3A2+FOXJ1+* cells, basal cells, club cells, and PNEC all neighbor TP and
 *SOX2^{high/low}CFTR+* cells.

We also confirm some of the spatial expression with immunofluorescence staining for the
 epithelial cell types as mentioned in response to #4. We also leveraged the Xenium spatial
 transcriptomics to confirm the presence and abundance of the expressed DEGs used to annotate
 each cell type in the epithelial, stromal and endothelial compartment (in **Figures 2E, 4D** and
 **Supplementary figs. 3A, 3C, 4D, 5A, and 5C** and **Supplementary Note 2**). For the epithelial
 population, we performed spatial and co-occurrence analyses to create a model of the developing
 airways in GW15 and GW18 fetal lungs (see figure below and now in main MS **Fig. 10**).

Here we provide a predicted model of the spatial localization of the different epithelial cell types
 in GW15 and GW18 which supports our inferred trajectory analysis of the origins and lineage

relationships of the epithelial cells. For example, we find budtip cells are closely associated with
tip cells in the budding regions of the airways which gives rise to *NKX2-1+SOX9+CFTR+* cells,
*SOX2^{low}CFTR+* cells, *SOX2^{high}CFTR+* cells and TP. These TP cells are closely surrounded by
differentiated PNEC in early development and later, ciliated precursors, proliferating
progenitors, and basal cells. We also find *NKX2-1+SOX9+CFTR+* clustered in the bifurcating
regions of the developing airways.

**Minor comments:**

1. “The authors sample from 19 different lungs, noting that one lung was sampled twice to
capture more regions of the lung. Where was the standard sample taken from a lung regionally
located, and what did it include? This description added to the methods would be helpful.”

For the tissue collection for scRNA-seq, we minced and isolated single cells from as much of the
lung tissues as possible. A small piece of later gestational tissue was collected, snap frozen for
storage. As we were unable to consistently retrieve the tracheal tissue, especially from smaller
lung tissues, we had decided to remove this tissue from our analyses. However, we recognize
that this may have resulted in the poor representation of the proximal lineages and indeed
prompted by the reviewers, we performed immunofluorescence staining of additional archived
fetal lung tissues (not the same samples), we did find basal cells in the large airways which were
not captured and represented in our sequencing data after. Therefore, we have clarified the
sampling of the tissue in the methods section. We have also included a description to the
methods (page 18, lines 715-719) to specify the sampling of the duplicated sample. It reads, “A
summary of tissues collected are listed in Supplementary Table 1. GW_18_1A and GW_18_1B
were isolated from the upper and lower airway regions from the same tissue. Tracheal tissues
were not collected for sequencing.”.

2. “Additionally, in table 1, GW_15_5_1 shows an increase from the number of sequenced
cells to the filtered number of cells. Does this represent experimental error? If so, why was this
sample used? 15_5 also has a large % increase of cells. Is this an error?”

We apologize for the mistake that was made and have created an updated Table that specifies the
correct number of cells sequenced for GW_15_5_1. There was not an issue with the sequencing
nor pre-processing of the dataset.

3. “There is a strong description of the cell composition of fetal lungs, with the authors also
claiming that epithelial subpopulations change over time. Table 3 reveals no discernible
patterns in the changes in proportions over gestational weeks. Since the sample sizes for
gestational weeks are so small ($n = 1$, $n = 2$, or $n = 3$), it should be noted that this could be due
to cells lost during processing, rather than an inherent biological phenomenon. It would also
be helpful to include a key in Figure 1E to show which proportions are varying – the
inclusion of these cell type data may make the figure more convincing than it currently is.
(What are the red, green and black boxes in Figure 1E? It is in the text, but not the figure
legend.) This is a confusing sentence: Cell type proportions of these cell types showed
changes in the presence of varying cell types (Supplementary Table 3).”

The reviewer raises a valid point in that indeed changes in proportions may reflect sample sizes,
sampling error, technical losses etc and given the low sample size for some of the tissues, it is
important to not conclude that changes are reflecting inherent biological outcomes. As such, we
have removed original Figure 1E which showed the UMAP of all the cell types across each
gestational time point where we had originally noticed what was perceived as changes in cluster
patterns. On a larger scale this may indeed be explained by the issues raised above. Under finer
resolution at looking at specific cell types, we do find certain cells develop over time while
others are lost at later time points (e.g. ciliated cells at the expense of PNEC) especially in the
epithelial compartment and we focus on these in the manuscript Figures 4-7.

We therefore have removed the table and sentence “Cell type proportions of these cell types
showed changes in the presence of varying cell types (**Supplementary Table 3**)” from the
manuscript.

4. “The first section of the supplementary note about the comparison with the other data sets
is not referred to in the main text. Although it is not clear what this adds over extended data 1 –
some of the figure panels seem to reproduced between extended data 1 and supplemental note
figure 1.”

We have removed duplicated data. As there are limits to the number of Extended Data (must
match the number of main figures) we are unable to include all of our analyses in either the main
figure or extended data. As such, we have moved additional analyses such as endothelial and
immune cell characterizations to **Supplementary Notes 2 & 3**, respectively (available online)
for those interested in our analyses of these cell clusters, especially since our focus was the
epithelial and stromal compartment and ending the atlas with the hPSC comparison.

5. “Supplementary note discussion of immune cells should refer to the new (previously
BioRxiv) ms characterizing the human fetal lung immune populations PMID: 38100545. For
example, this published ms also finds pDC (which is described in this ms as a newly-
identified cell type in fetal lung), and there are other similarities between the analyses of the
immune system here and in the recent publication.”

We thank the reviewer for specifically bringing this to our attention as we had missed this
BioRxiv manuscript characterizing the immune cells. We are excited about the similarities
observed and have properly referenced their work in where we have found common findings. In
our trajectory analysis of epithelial cell fates, we find the expression of interleukin pathways
(Figure 6). This may indicate signaling from immune cells as observed in Barnes et al. and have
indicated this in the manuscript text.

6. “We suggest similar edits to the text (as minor point 3 above) to the effect that sampling error
cannot be excluded as a source of changes in cell distribution from sample-to-sample when
discussing the varying proportion of fibroblasts over the fetal period in Figure 2B.”

We agree that changes in cell proportions may reflect sampling error and have included a
statement in the Methods section to clarify our tissue collection limitation (e.g. our dataset
excludes cells from tracheal tissue) and sampling error may have skewed our dataset. We have
mentioned this as a possible explanation for differences in cell type overlap between our dataset
and previous datasets on pages 4-5, lines 153-156.

7. “When describing the dynamic fetal lung epithelium, the CFTR+ subtypes are described
as novel. This is not true. While the analyses that the authors do on this cell population are novel
and very interesting, the cell population itself (SCGB3A2+SFTB+CFTR+) has previously been
identified by Conchola et al. This is referenced later in the manuscript, but it should be
referenced at the first mention of this cell type.”

We thank the reviewer for this comment and have ensured the work by Conchola et al. is put at
the forefront at first mention of the SCGB3A2+SFTPB+CFTR+ or TP population. We have
acknowledged this on page 4, line 114-115.

8. “The discussion about SOX2/SOX9 co-expression in the developing human bud tips
could do with mentioning that previous work relied on antibody staining, and protein and mRNA
distributions can be different. Meaning that this work, and previous immunofluorescent studies,
may well both be correct.”

We thank the reviewer for the suggestion here and have added references to reflect that protein
and mRNA expression patterns may indeed differ but be correct as mRNA does not always
translate to protein and protein expression may reflect longevity of protein stability before
degradation.

“While *SOX2* and *SOX9* co-expressing cells have previously been identified in double positive
distal tip lung progenitors, the expression of *SOX2* in the budtip progenitors in our dataset is
relatively low compared to *SOX9* and suggests the captured budtip progenitors are similar to
previously found by Nikolic et al. (PMID: 28665271), Miller et al (PMID: 29249664), and
Danopoulos et al, (PMID: 28971977). Differences in mRNA and protein expression patterns are
possible as there is not always a correlation under static observations since the turnover of
mRNA and protein may be different (ref PMID: 26053859).”

9. “Fig. 3F does not denote the difference between high and low SOX2+ expression strongly –
there should be some quantitation added.”

We have stained fetal lung tissues and now performed quantification of SOX2^{high} vs SOX2^{low}
shown in new **Supplementary Fig. 4C**. Each dot represents 6 random fields obtained and
quantification was determined using Velocity software to segment areas of high vs low
expression based on fluorescence intensity. Using this we have demarcated areas of high (green)
and low (blue) SOX2-expressing cells across gestational weeks and within airways. However,
there was no statistically significant difference based on student’s t-test between the proportion
of high and low SOX2 within each GW.

10. “In Fig. 4, TP is never denoted in the text but is used on the figure with no explanation of the
abbreviation. The abbreviation should be stated with its explicit definition in the text.
Additionally, in the text it is stated that basal cells were not captured in early stages of
development. Is it possible Fig. 4g does not represent a late stage “acquired” developmental
plasticity – the basal cells simply were not captured in the earlier timepoints?”

We have added the abbreviation of TP and its explanation in the text on page 8, line 315: “The
*SCGB3A2*+*SFTPB*+*CFTR*+ cells or referred to triple positive (TP) herein,...”.

Our dataset does not include tracheal tissues and therefore we recognize our dataset may not
 reflect a good proportion of proximal cells to capture sufficient/significant numbers of basal
 cells. This may explain why we do not observe basal cells in our early stage samples. To confirm
 this, we performed immunofluorescence staining of archived GW12 and 15 fetal lung tissues and
 found a greater proportion of basal cells (marked by dP63 in yellow) to be present in large
 airways relative to the smaller airway (See images below, scale = 100 microns). Therefore, we
 have revised our manuscript to state the limitation of our scRNA-seq dataset in not capturing a
 good representation of large, upper airways. It reads, “This may represent insufficient sampling
 of the larger airways such as tracheal tissues (which was not collected for sequencing).” as it has
 been done in other datasets (ref 36 for Miller et al, ref 10 for He et al) on **page 10, lines 365-376**.

11. “In Fig. 6d, the pearson correlation coefficients are extremely similar aNRt early stages,
which would make hPSC fetal lung cells and organoids both partially representative for early
stages. It should not be claimed that the organoids are a significantly worse early-stage model.
However, the data do show that the hPSC organoids would be the better choice for modeling
later stages of development.”

We agree that hPSC organoids are not a significantly worse early-stage model. Rather, we
intended to say fetal lung cells captured an earlier stage of the fetal lung. We re-analyzed this
data using Spearman correlation analysis (<GW12, new **Figure 8F**). Proliferating progenitors
can also be seen in hPSC Organoids cluster 1 and 3 in **Figure 8E** (which also have a low hPSC
organoids pseudotime, suggesting they represent an earlier developmental time point, in **Figure**
**9**). We have revised the description of the two hPSC-derived fetal cells and organoids to reflect
what they represent, the overlap and that the organoids captured some of the differentiated cells
observed in later stages of development. The latter can be a better model to study lineage
development of those cells. The description is on **pages 14-15, lines 583-618** and describes new
**Figure 8**, which shows hPSC cells analyzed clustered independent of integration with the
primary fetal tissues and computing differentially expressed genes (DEGs) compared to the
DEGs in the primary fetal sample. We then show in new **Figure 9** the integrated hPSC dataset
with the primary fetal lung sample and found the same outcome where hPSC fetal lung cells
mostly captured proliferating progenitor cells observed in early fetal stage of development,
whereas hPSC organoids reflect later stages.

**Response to Reviewer #2:**

In the current manuscript, Quach and Farrell carry out a single-cell sequencing atlas from 19
healthy human fetal lung tissues from gestational weeks 10-19 and they identified at least 58
unique cell types/states contributing to the developing lung. Overall, the manuscript represents
an interesting resource that will be well utilized in the field. The authors describe cell
populations and use computational approaches to infer lineage trajectories, transcription factors
and cell signaling during differentiation. I have listed several comments below, organized into
‘major’ and ‘minor’ categories. Overall, it would be nice if the authors could test some of the
hypotheses generated by the computational approaches in order to make new biological
observations.

We thank Reviewer 2 for the very insightful comments. While we agree that experimental
validation of some of our inferred trajectories would definitely be of interest, our intent of this
manuscript was to keep this a reference atlas paper. Importantly, our inferred trajectories have
not only confirmed some previous experimentally validated lineage relationships by He et al and

Conchola et al, but we also find new trajectories (possibly due to our larger dataset). This
dataset, if combined with the published datasets, can provide a rich and informative collection
that may reveal even more lineage relationships and how other fetal lungs (ie. immune,
endothelium, Schwann cells) support the development of the epithelia/stroma and vice versa. In
this revision, we also provide high resolution spatial resolution using the 10X' Xenium platform
and custom panel of 300 genes curated from our scRNA-seq dataset combined with the 10X lung
panel. This enables us to use spatial information to determine cell proximities and better resolve
cell-cell signaling that may drive cell fate decisions in TP cells. Moreover, an additional
objective of this work was to assess the validity of hPSC-derived cultures as these may be used
in future studies to track fetal epithelial lineage development through the fetal stages and beyond
(with differentiation). We appreciate Reviewer's 2 comments and are looking to model the
beautiful work by Conchola et al (ref 13) where lineage tracing studies using both primary fetal
tissue explants and hPSC-derived models and focus on a specific cell lineage.

**Major comments:**

1. "Regarding Figure 2, the authors state: "Further investigation into differentially
expressed transcription factors in these cells showed uniquely higher expression of SOX2 and
NRD1". This is quite surprising, given that several other studies have interrogated SOX2 in the
context of human fetal lung development, both using sequencing and protein based approaches,
and have not described SOX2 in mesenchyme. The authors should validate their findings using
other approaches (FISH, IF) to show co-expression of SOX2 and mesenchyme specific markers,
using high resolution imaging."

We thank the reviewer for noticing this statement, which somehow evaded our detection in the
many internal review processes. This statement is an error, and we have now removed it from the
manuscript as SOX2 is not expressed in the mesenchyme. We confirmed this using
immunofluorescence staining, which showed distinct expression of SOX2 (marked in yellow) in
the airways but not in the mesenchyme (marked by vimentin in green) of the fetal lungs. The
white hashed boxes in the left column images indicate regions where high-magnification images
were taken, corresponding to the detailed images in the adjacent column. Scale = 25 microns.

2. “Regarding the fetal epithelium, the authors state: “Mature ciliated cells and
 SCGB3A2+FOXJ1+ cells emerged around GW14 whereas basal cells and SMG secretory cells
 emerged around GW15, but these latter cell types represented only a very small fraction (~0.2-
 1% and ~1-5%) of the total epithelial cell population”. This statement goes against multiple
 published studies, including Miller and Yu et al., Dev Cell (2020), He and Lim et al., Cell
 (2022), which show that basal cells are present as early as 9 PCW. Could the data in this study
 reflect poor representation of some regions of the lungs, for example, more proximal airways and
 trachea are not adequately represented in the data set shown here? The authors should carefully
 evaluate their claims in the context of these previously published studies.”

We agree with the reviewer’s concerns and have since updated the methods section (page 18,
 lines 723-724) to clarify that the tracheal regions of the collected tissues were not included in our
 analysis. We have also clarified this in the results section of the manuscript to explicitly say,
 “This may represent insufficient sampling of the larger airways such as tracheal tissues (which

was not collected for sequencing).” Please refer to our response to “Reviewer 1: Minor Comment
10”

3. “The authors state: “Interestingly, while SOX2 and SOX9 co-expressing cells have
previously been identified in double positive distal tip lung progenitors, the expression of SOX2
in the budtip progenitors in our dataset is relatively low compared to SOX9”. This is not
surprising, and SOX9-high, SOX2-low expression (both RNA and protein) in human bud tip
progenitors has been reported previously by the Al-Alam (PMID: 28971977), Rawlins (PMID:
28665271) and Spence (PMID: 29249664) laboratories. This statement should be revised so as to
not sound like this finding is new and/or surprising, and prior literature should be properly
cited.”

We appreciate the reviewer’s comment and have revised this sentence in the manuscript to
acknowledge the same finding by Al-Alam (Danopoulos et al, PMID: 28971977), Rawlins
(Nikolic et al., PMID: 28665271), and Spence (Miller et al., PMID: 29249664). The sentence on
page 8, lines 274-275, now read, “Expression of *SOX2* in the budtip progenitors in our dataset is
relatively low compared to *SOX9* which corroborates with previous findings^{35,36,34}.”

Differences in mRNA and protein expression patterns are possible as there is not always a
correlation under static observations since the turnover of mRNA and protein may be different
(PMID: 26053859).

4. “The author claim to identify novel SOX9-low, SOX2-low cell populations, stating:
“Instead, cells that appeared to express both SOX2 and SOX9 at relatively equal levels included
SOX2lowCFTR+ cells, PTEN+STAT3+ cells, basal cells, and the proliferating progenitors.”
However, there is no high-resolution validation of these populations. Are SOX9-low cells
actually expressing protein, or is this result due to low levels of mRNA that are captured as cells
transition/differentiate from the bud tip progenitor state into a new differentiated state?”

We have recreated the dotplot for *CFTR*, *SOX2* and *SOX9* which is now in new **Figure 4C** using
a two-tone colour to show low (blue hues) and high (red hues) expressions. Here, we can better
see that *SOX2* and *SOX9* co-expressions which are not captured. *SOX2*^{high} and *SOX2*^{low} is better
visualized by protein analysis using immunofluorescence staining (**Supplementary Fig. 4C**).
However, we see less of this broad expression range with *SOX9*. Rather, cells that express
abundantly higher levels of *SOX9* include budtip and Tip cells and less so *NKX2-*
*I+SOX9+CFTR+*. It is hard to determine based on the “low” expression of *SOX9* that it
indicates these cells are transitioning/differentiating without experimental lineage tracing

validation. However, our inferred trajectory predictions (**Figure. 6F-G**) and Xenium spatial
 localization suggest that there is a relationship between *SOX9*+ cells and spatial expression
 suggests these cells all reside close to one another in the bud or bud-adjacent region Tip and next
 to stalk regions. We therefore predict that *NKX2-1+SOX9+CFTR+*, while they do express low
 levels of *SOX9*, may represent a transitional state from Tip cells as they differentiate or acquire
 stalk identity. In assessing their DEGs, these cells are distinct from Tip and *SOX2*+ cells, but
 have similarities with both (see DEG overlap plot below, new **Supplementary Fig. 6F**), further
 supporting their lineage relationship. Immunostaining for SOX9 in **Figure 4E** shows SOX9
 protein is abundant in the developing airways.

5. “Given the fact that CFTR expression is being used as an important defining gene for
 several cell populations/states, it is unclear why CFTR expression data is relegated to
 supplemental figure 3. Similarly, supplemental figure 3C would be very helpful for defining the
 cell populations in Figure 3A, given that many of the canonical, lineage defining markers are
 shown in Supp Fig 3C. Figure 3B and Supplemental Figure 3C should be combined into one
 comprehensive heatmap and/or bubble plot, which includes canonical lineage defining genes,
 alongside all genes used to define cells in this study. It would be helpful for gene names to be
 easily discernable (and not squished together as is the case in Supp Fig 3C). For example, is

FOXJ1 actually ever plotted? This seems essential since the authors use FOXJ1 to define a
“SCGB3A2+ FOXJ1+” cell population.”

We have moved all *CFTR* expression data to the main manuscript figures including the dotplot
showing *SOX2*, *SOX9* and *CFTR* expression to new **Figure 4C**. We have also updated the
heatmap in new **Figure 4B** where we first describe the epithelial clusters to include canonical
genes as well as the top DEGs associated with each cluster. As the epithelial cluster is rather
large and has many cell subtypes, we have kept some additional data in new **Supplementary**
**Fig. 3C**. Specifically, the proportion figure (new **Supplementary Fig. 4A**), the SCENIC
heatmap of differentially active regulons for transcription factor activity (new **Supplementary**
**Fig. 4B**), high resolution spatial analysis of *TP63*, *SCGB3A2* and *FOXJ1* in GW15 and GW18
lung tissue (new **Supplementary Fig. 4C**), and additional immunostaining for epithelial subtype
markers (*TP63*, *SCGB3A2* and *FOXJ1*, new **Supplementary Fig. 4D**) for the different
gestational time points.

We have also carefully re-generated new figures to increase font size and consistency of all the
sub-figures and hope this version is not readable.

6. “Based on Supplemental Figure 3B, it seems that some level of *CFTR* is expressed in
every cell population, and the data shown contradicts the statement “A novel finding was the
identification of four epithelial cell types that expressed relatively high levels of *CFTR*.””

We understand how “Extended Figure 3B”, now in the main figure “new **Figure 4C**” which
showed a dotplot of *CFTR*, *SOX2* and *SOX9* in all the epithelial cells, suggested some level of
*CFTR* in all cells. It is important to note here that the expression of these genes when shown in
this way show relative expression compared to the rest of the cell types. To better clarify the
differences in *CFTR* expression, we have replaced the dotplot with a two-tone plot to better show
low (blue hues) vs high (red hues) expression. The figure now clearly shows that the 4 progenitor
cell populations: *SCGB3A2+SFTPB+CFTR+*, *SOX2^{high}CFTR+*, *SOX2^{low}CFTR+* and *NKX2-*
*I+SOX9+CFTR+* express relatively higher levels of *CFTR* than the other cell types.

7. “The authors show *CFTR* localization using antibodies, but *CFTR* antibodies are
notoriously unreliable. The authors should validate the specificity of this reagent and possibly
confirm localization using a complimentary approach such as FISH.”

We appreciate the reviewer’s concerns regarding the reliability of commercial *CFTR* antibodies
and the controversies surrounding them. To address this, we carefully selected the MAB1660

clone #13-1 antibody from R&D Systems to target CFTR. Yeager et al. (PMID: 25453871)
conducted a comprehensive study comparing this CFTR antibody with 16 others, using a variety
of samples, including CF and non-CF lung tissue, as well as primary human lung epithelial cell
lines. Through their extensive immunofluorescence staining experiment, the authors observed the
R domain-specific antibody, clone #13-1, consistently showed good apical membrane CFTR
staining in non-CF cells, and staining was not seen in most F508del CFTR cells, confirming its
specificity.

Additionally, to validate our findings further, we performed Xenium analysis to visualize CFTR
localization on our fetal tissue. This provided high-resolution spatial transcriptomic images,
complementary to our immunofluorescence staining, further supporting the reliability of our
antibody choice.

8. “The authors state: “Transcription factors enriched in early AT2-like cells included ETV1
and HNF1B, as previously described (ref. 52) as well as SOX9 and FOXP2”. However, Ref. 52
does not discuss ETV1 at all, and mentions HNF1, but not HNF1B. Moreover, it is unclear what
the authors mean by “early AT2-like cells”, since Ref. 52 carries out “...epigenomic profiling of
human AEC during differentiation from AT2 to AT1-like cells. AT2 cells were extracted from
explant donor lungs that had no prior evidence of chronic lung disease and allowed to
differentiate into AT1-like cells in vitro over the course of 6 days utilizing well-established
protocols”. Thus, all studies in ref. 52 are carried out in adult human AT2 cells.”

We appreciate the reviewer’s concern about the references used to relate the transcription factors
enriched in “early AT2-like” which we now call “Tip” cells. To clarify, reference 52 (Zhou et
al., PMID: 34922464), specifies *HNF1B* as the transcription factor expressed in AT2 cells in Fig.
4C. Nonetheless, we agree with the reviewer’s concern as the studies are done in adult human
AT2 cells and have since removed the reference. We have also added the reference that discusses
ETV1 as a homolog for *ETV5* by Herriges et al (page 9, line 354).

We annotated these cells as AT2-like based on the differentially expressed genes (DEGs) that
appeared to separate these cells from the classical budtip cells (Figure 4B). While there are genes
commonly expressed in budtip cells and AT2-like cells, genes that are differentially enriched in
budtip cells such as *SFTPC* which we used Xenium to confirm the relationship between Budtip
and now “Tip” cells. Also suggested by Reviewer #1, we show that the Tip cells are discrete
from the AT1/AT2 cells, and earl/mid/late Tip cells by Peng He et al. Please see response to
Reviewer #1 response #5.

9. “The authors state: “Several trajectories converged onto SCGB3A2+SFTPb+CFTR+ in
the “mid” gestational tissues. These include cell sources from SOX2^{low}CFTR+ cells,
SOX2^{high}CFTR+, stromal-like cell 2, proliferating progenitors and basal cells. A strong
connection (thick arrow line) between NKX2-1+SOX9+CFTR+ and SOX2^{high}CFTR+ cells and
the latter with SCGB3A2+SFTPb+CFTR+ were observed in all timepoints, suggesting these
cells are developmentally related and stayed consistent through the 10 weeks.” Some
experimental evidence to connect these computationally inferred trajectories would be ideal.”

The reviewer is indeed correct in that there is a “stronger” (thicker arrow line) connecting *NKX2-*
*1+SOX9+CFTR+* with *SOX2^{high}CFTR+*, and then the latter with *SCGB3A2+SFTPb+CFTR+*
(TP) (new **Figure 6E**). We also agree that experimental evidence to connect these inferred
trajectories would be great and is planned in our upcoming studies leveraging both fetal lung
organoids as elegantly performed by Conchola et al (ref 13) to validate TP differentiation, along
with our hPSC-derived models, which we have now shown does generate these cell types. These
models can be used to validate the relationship between *SOX2^{high}CFTR+* and *SOX2^{low}CFTR+*
cells with TP and then TP with PNEC, ciliated and basal/club cells (as observed in later
gestation) and beyond with the hPSC differentiation towards “mature” cell cultures. We hope to
share this in future publications.

10. “Figure 4A labels colors the clusters, and then has a key that denotes which cluster
corresponds to which color. It is impossible for the reader to discern between some of the shades
of similar colors. Then, Figure 4C and D use numbers to label the clusters. The numbers are far
easier to identify; however a numbered key is not provided. This figure is therefore very
frustrating to try and connect with the text.”

We thank the reviewer for noting these nuances which can be frustrating to connect. We have
updated the UMAPs to include numbers embedded on the UMAPs and in the corresponding
legend. We have made it a point to keep the cluster colours the same in all UMAPs pertaining to
that cell type with the exception of Figure 1 (UMAP of ALL cell types). Since **Figure 1** is a
large UMAP of all the cell types/states, we have chosen a colour scheme to better highlight
individual cell types/states than the one previously used, which in hindsight, was cumbersome to
discern.

11. “The authors state: “Within Visium spots, spatial proximity of PNEC and
SCGB3A2+SFTPb+CFTR+ cells were identified at week 15, while spatial proximity of mature
ciliated cells, SCGB3A2+FOXJ1+ cells, and SCGB3A2+SFTPb+CFTR+ cells were found at

616 week 18 (Fig. 3C and 5E).” However, at week 15, it seemed that the co-occurrence was much
more common for several other cell types, including lipofibroblast precursors, which are not
even of epithelial origin. This seems odd.”

A caveat of the Visium spots is the broad capture area for each spot (55 micron in diameter).
Specifically, the resolution of the Visium spots is not single cell level and therefore we often find
other cell types (ie. stromal and not just epithelial) within the spots, so this is not surprising. This
is a technical limitation of Visium. We acknowledge that the resolution may confound the
analysis and have opted to use 10X Xenium to replace many of the Visium images throughout
the manuscript, which provides much higher resolution to address the proximity of these cell
types. In the new **Figure 5A**, we show TP cells in close proximity to mature ciliated cells, basal
cells, PNEC, *SCGB3A2+FOXJ1+*, and *SOX2^{high}CFTR+* cells. With Xenium, only at increasing
distances away from these cells do we start to see stromal cell types such as airway fibroblast
progenitors, airway SMCs, and lipofibroblast precursors (new **Figure 5B**). These changes are
now updated in the manuscript where we discuss each cell type. We also compare the co-
occurrence analysis of the cell types from Visium to Xenium (in new **Figure 5A and 5B**). While
Xenium provides high single cell resolution of the spatial expression of genes, it is not whole
transcriptome (like Visium). Leveraging the results from Visium and Xenium, we are more
confident that these cell types listed are neighboring, and that the stromal cells showing up as
neighboring in Visium are just the nearest stromal cells, and Xenium allows us to examine their
proximity in detail.

12. “Inferred signaling networks are interesting. It would be great if the authors tested some
of the hypotheses generated in the computational data. For example, does NOTCH signaling play
a role in PNEC differentiation, as discussed in the text?”

Testing the inferred signaling networks is indeed a next step in our continued study of the
developing fetal lung. To keep this as a cell atlas for which we would like to share with the
broader academic community, we have chosen to focus solely on describing the large dataset we
have collected and is enabling us to perform these inferred analyses. However, it is interesting to
note that previously published work has shown that hPSC differentiation to PNECs is dependant
on NOTCH inhibition (PMID: 32380423). We showed that there is a correlation between PNEC
numbers and NOTCH signalling to TP (**Supplementary note 5C**), the presumed source of
PNEC differentiations. We have included this in the discussion (**pages 16-17, lines 674-678**).

13. “It isn’t clear why iPSC-differentiated lung cells were immediately integrated with
human fetal lung data? It would be much more compelling if (1) iPSC data was analyzed on its
own and canonical lung markers were shown; (2) iPSC data was label transferred onto human

fetal data (i.e. human fetal data is used as a search space, iPSC data is computationally projected
onto the fetal data based on cell similarity).”

As recommended, we have separately analyzed the hPSC cells on their own by clustering,
computing differentially expressed genes (DEGs), and comparing to the DEGs in the primary
fetal sample which express many of the canonical lung markers. We confirm our earlier analysis
in which the DEGs of hPSC organoids mostly overlap with basal, club, *SCGB3A2+FOXP1+*, TP,
*SOX2+CFTR+* cells, while the DEGs of hPSC fetal lung cells most strongly overlap with
proliferating progenitors and budtip progenitors. This is shown in new **Figure 8 (B-C and D,E)**.
We also performed a Spearman correlation and showed that the hPSC-derived fetal lung cells
have better correlation to early gestational fetal lung tissues (<GW12). Meanwhile, the hPSC-
derived organoids have better correlation to later gestational fetal lung tissues(>GW16) (new
**Figure 8F**).

In new **Figure 9**, we show our original analysis with continuous cell type scores based on
expression of marker genes (ie. where we integrate the iPSC data onto the primary fetal lung data
to identify cell similarity).. More importantly, we performed trajectory analysis and showed the
change in these celltype scores along the trajectories increased for the more differentiated cell
types (e.g. basal, and *CFTR+* cell types).

Forcing a particular discrete label onto the hPSC generated cells from the fetal cell type may not
be appropriate because none of the hPSC clusters fully match the fetal cell clusters, and not all of
the genes seen in the fetal samples are expressed in the hPSC general cells.

**Minor comments:**

1. “Figure panels are stretched and compressed in an odd manner. Compare Figure 1D and
E. This is the same embedding, but stretched/compressed differently.”

We have fixed the stretched/compressed images and resized all the subfigures to make
them consistent, at least within each figure.

2. “Fonts in many panels appear to be weirdly stretched. Figure 1C is a good example of
this.”

We have now edited all the figure panels and tried to harmonize all font types and size for
consistency throughout the paper.

3. “Many different fonts are used in each figure. In Figure 1, it seems that at least 6 different
fonts are used in different panels.”

Please see answer above.

Response to Reviewer #3:

1. The authors present a fourth scRNAseq dataset addressing the differentiation of cell types in the embryonic human lung from w10 until gestational week 19 (w19). The previous published datasets of He et al and Sountoulidis et al, were published about a year ago and the 3rd paper (Can et a) about half a year ago. The sc RNA seq data of this manuscript are of high quality and are consistent with the 2 first previous papers. The authors align their data with the 2 first previously papers and suggest some new clusters epithelial cells. in Fig 1 G. There are however more clusters published by the other papers that are not detected here Figure 1D. The previous papers claim to detect about 80 cell states. It is important that the authors include also these in figure 1 G and provide an explanation for this discrepancy in the main part of the paper.

We have indeed captured 58 different cell types/states using our method of cell type resolution
(See response #2 to Reviewer #1), which does not completely align with the total cell type/state
count as previous fetal atlases from He et al (ref 10), Sountoulidis et al (ref 11) and Cao et al (ref
12). It is important to note that there are several differences in 1. the collection range of our
dataset, which includes gestational/developmental time points not captured in the others; 2.
Sampling of tissues for sequencing can vary, even though we aim to harvest consistent areas of
the lung each time for single cell processing. Our dataset does not include tracheal tissue and
therefore cells that are unique/abundant in the large or upper airways are not adequately
represented; 3. Our dataset does not capture earlier developmental time points that the other three
datasets includes (ie. our earliest tissue is gestational week 10, which is approximately equivalent
to PCW 8.0). As such, we expect some non-overlap in the cell types/states captured. 4. We have
not manually dissected the tissues based on “proximal” and “distal” regions as in He et al., and
there cannot cluster the cells based on regional localization. Instead, we opted for the unbiased,
blinded approach to annotating our cell clusters and performed Query analyses to determine
whether there are shared cell types/states captured in our dataset. 5. And finally, we are unsure
how tissue processing varies across labs which may impact the integrity of the sequenced data.
We collect “freshly” isolated lung tissues (ie. shortly after termination procedures) and process
the sample for single cell library prep on the same day. The tissues are carefully stored and
transferred between sites on ice to limit tissue degradation. In addition, differences in cluster
annotation based on DEGs may differ and this may be impacted by the number of cells we
captured.

We have now included a new **Figure 1F** (which includes Cao et al dataset, as per reviewer #1's
suggestion) and show where our dataset overlays on the combined datasets by He, Sountoulidis,
and Cao et al. We also have updated **Figure 1G** to show the outstanding cell types that our
dataset has captured but not identified in the other datasets. It is important to note that in our
previous comparison, we had arbitrarily set a cell number limit of 10 in which, cell clusters with
less than 10 cells are not considered "captured" in our dataset. We have now removed that limit
and therefore even if there is 1 cell shared in a cell type/state between our datasets, it is now
considered "captured".

2. "They provide a relevant short discussion in supplementary note 1 but this becomes lost
in the organisation of their writing. Ideally the data of the first two papers and this one should be
integrated into a as single dataset and provide a new consensus clustering , which would be very
useful for the human lung development field."

We appreciate the reviewer's comment and are currently in the progress of working with the
Human Cell Atlas to compare the cell states identified in this paper with previous papers.
Integrating and comparing our dataset is beyond the scope of this study for which we were
aiming to provide a fetal lung cell atlas which now adds to existing atlases. Part of integrating
single cell datasets is coming to a consensus on clustering annotations and this requires extensive
multi-lab collaborations with the original authors of the published dataset, again beyond the
scope of this paper. However, we are working towards the integration of all datasets and
contributing to the HCA moving forward.

To add to the analysis in comparing our dataset with the previously published datasets, we have
performed reverse queries (new **Supplementary Note 1**) using the Seurat *MapQuery* function in
which we map our datasets (query dataset) onto the published dataset by He, Cao and
Sountoulidis (reference datasets). Here we show that our dataset does capture many of the cell
types reported in He et al, Sountoulidis et al, and Cao et al but not all. This may be explained by
the fact that some of these datasets included datasets of earlier fetal lung tissues and the
separation of proximal and distal analysis of the lung which our dataset does not include. We
have described these additional analyses in the manuscript and detailed these additional analyses
in the methods section (Pages 19-20, lines 777-787).

3. "They present a combined Umap projection in Figure 1F but its is not at all clear to me
how they derived it and what kind of parameters they used for the potential integration or
correlation. a more detailed description of the methods is necessary."

We have updated the methods to include the integration analysis done for Figure 1F:
“Integration analysis with publically available datasets was done using Seurat’s rPCA method
to find integration anchors using default parameters, and subsequent integration using
IntegrateData. Standard workflow (as above) was done to process the integrated assay and
UMAP visualization.” in (page 19, lines 771-774)

4. “The authors focus their analysis on the gene expression trajectories leading to fibroblast
and epithelial cell differentiation. I am confused by the PAGA plot in figure 2F . Are the arrows
of equal thickness indicating equal probabilities, or is it a random selection of line thickness?
They look very similar. The Velo-plot in fig. 1e does not show much continuity between cluster
4 and 5. Are the PAGA plot and veloplots consistent?”

The line thickness corresponds to transition weights between clusters. Because we exclude
chondrocytes and smooth muscle cells (SMC) from the trajectory analysis, we have recomputed
the UMAP embedding for this subset. Thank you for pointing out the continuity issues of the
velocity plots. We have now included velocities embedded on the UMAP that visually align with
the PAGA (new **Figure 3**).

5. “Overall these results are consistent with previously published mapping efforts of
Sountoulidis et al but lack cellular resolution. Interestingly the authors define a spatial
relationship of lipofibroblasts with epithelial and endothelial cells at the tips of the branches.”

We agree our previous spatial resolution using 10X’ Visium was not high with the best showing
“spots” showing co-occurrence of multiple cell types in the area. Nonetheless, we did find
similar mapping efforts as Sountoulidis et al. To show higher cellular resolution, we have now
performed Xenium spatial expression of the top DEGs using a custom panel of genes aimed at
capturing mainly all our epithelial and stromal cell types. Through this, we are able to hone in on
cell proximities with statistical significance associated with TP cells (our focus of this paper) and
includes $SOX2^{high}CFTR+$ cells, PNEC, club cells, and basal cells in GW15 lungs; and
lipofibroblast precursors, $SCGB3A2+FOXJ1+$, airway SMCs, and mature ciliated cells in GW18
lung (**Figure 7E**).

6. “The analyst would be better understood if they provide information on the numbers of in
vitro grown cells , the number of detected genes and how the cells clustered by themselves. The
authors make an extensive description of potential regulators in the embryonic lung and would
be interesting to know if specifically these are also detected in the invitro cultured cells.”

We have included additional details regarding the number of cell lines used, and the number of
 detected genes in the methods section (Page 23, lines 906-907). It reads, “A total of 2493 and
 2450 cells of fetal lung cells and organoids were sequenced, and 23,968 and 28,648 genes were
 captured respectively.”

We have separately clustered the hPSC fetal lung cells and hPSC organoids (new Figure 8). We
 compare the DEG of these clusters to the DEG of the fetal lung clusters to characterize them in
 relation to the fetal tissues. We confirm our earlier analysis in which the DEGs of hPSC
 organoids mostly overlap with basal, club, *SCGB3A2+FOXJ1+*, TP, *SOX2+CFTR+* cells, while
 the DEGs of hPSC fetal lung cells most strongly overlap with proliferating progenitors and
 budtip progenitors. This is shown in new Figure 8 (B-C and D,E).

The in vitro cultured system is devoid of co-developing stromal cells as the directed
 differentiation is intended to generate a cell type (epithelial), so many of stromal source cell
 types identified for signaling will not be present. However, in each step of the differentiation, the
 culture media is supplemented with factors that are presumably stromal cell-derived (ie. FGF10)
 which signals to the developing epithelium (PMID: 35025140,
 PMID: 22922672, PMID: 25654755). Indeed, we find FGF receptors are expressed in the hPSC
 derived cells but not ligands (see figure below). Similarity, WNT receptors are expressed but not
 the ligands. In the case of NOTCH, there is expression of the ligands, which might be expected
 since we see potential signaling from *SOX2^{high}CFTR+* epithelial cells and not just stromal.

7. “An overall suggestion for improvement is to provide specific values for the detection
 and fold enrichment of the DEGs listed in the supplementary table and . Also the panels showing
 visum data are often lacking scale bars.”

We thank the reviewer for this suggestion and have included the specific information in
 Supplementary Table 2, we have provided these additional values in separate excel sheets for
 each cell type.

Where relevant, we have included the scale bars for all images.

REVIEWERS' COMMENTS

Reviewer #1 (Remarks to the Author):

The authors have done an excellent job of responding to all the reviewers' essential comments. We agree that the mechanistic analyses are out of the scope of this publication.

2 minor points:

1. Different antibodies for CFTR are listed in the methods section and the rebuttal letter. Please confirm that the correct one is listed in the ms.

2. A figure legend for figure 10 seems to be missing.

Reviewer #2 (Remarks to the Author):

In their revised manuscript, Quach and Farrell et al., carried out a major overhaul of the text and figures based on extensive feedback. While the scope of the manuscript is limited by its exploratory and hypothesis-generating nature, it does represent a large dataset that builds on prior resources by including a large number of new fetal lung data sets and spatial transcriptomic data that will be very useful for the field. With the extensive revisions carried out in the current version, the reviewers have satisfied the majority of my original critiques.

Reviewer #3 (Remarks to the Author):

The authors made substantial efforts to address my comments and have improved the manuscript. I believe that the paper is interesting and useful in the emerging field of human lung development. I found a mistake that the authors need to correct. They write in the new text "A total of 2493 and 2450 cells of fetal lung cells and organoids were sequenced, and 23,968 and 28,648 genes were captured respectively." It must be wrong that they detected 25000 genes. Do they mean UMIs?

Reviewer #4 (Remarks to the Author):

First and foremost, we wish to express our sincerest gratitude to the reviewers for their
constructive feedback and suggestions which led to the significant improvement of our revised
manuscript. We thank the reviewers for noting the minor issues for us to address and have
ensured these are fixed in the final version of the manuscript.

**Response to Reviewer #1:**

1. Different antibodies for CFTR are listed in the methods section and the rebuttal letter. Please
confirm that the correct one is listed in the ms.

We apologize for any miscommunication. We only used 1 CFTR antibody MAB1660 clone #13-
1 antibody from R&D Systems in our figures which has been previously shown to be a reliable
antibody for CFTR immunostaining. We confirm only this antibody is listed in the manuscript.
For clarity, we have revised the rebuttal:

*“We appreciate the reviewer’s concerns regarding the reliability of commercial CFTR*
*antibodies and the controversies surrounding them. To address this, we carefully selected the*
*MAB1660 clone #13-1 antibody from R&D Systems to target CFTR in our immunofluorescence*
*staining. Yeager et al. (PMID: 25453871) conducted a comprehensive study comparing this*
*MAB1660 clone #13-1 antibody with 16 others, using a variety of samples, including CF and*
*non-CF lung tissue, as well as primary human lung epithelial cell lines. Through their extensive*
*immunofluorescence staining experiment, the authors observed the MAB1660 clone #13-1,*
*consistently showed good apical membrane CFTR staining in non-CF cells, and staining was not*
*seen in most F508del CFTR cells, confirming its specificity. Additionally, staining conducted by*
*our lab on CF and non-CF cell lines (unpublished) also showed similar CFTR staining results to*
*Yeager et al. Therefore, MAB1660 clone #13-1 was selected for immunofluorescence staining.*

*Additionally, to validate our findings further, we performed Xenium analysis to visualize CFTR*
*localization on our fetal tissue. This provided high-resolution spatial transcriptomic images,*
*complementary to our immunofluorescence staining, further supporting the reliability of our*
*antibody choice.”*

2. A figure legend for figure 10 seems to be missing.

We sincerely apologize for this oversight and have included a figure legend for the graphical
summary of our findings in Fig. 10.

**Response to Reviewer #3:**

The authors made substantial efforts to address my comments and have improved the

manuscript. I believe that the paper is interesting and useful in the emerging field of human lung
development. I found a mistake that the authors need to correct. They write in the new text "A
total of 2493 and 2450 cells of fetal lung cells and organoids were sequenced, and 23,968 and
28,648 genes were captured respectively." It must be wrong that they detected 25000 genes. Do
they mean UMIs?

This is indeed a mistake and we are grateful that Reviewer #3 found this. We have made the
correction in the manuscript on page 21.
